# Generalized Orders of Magnitude for Scalable, Parallel, High-Dynamic-Range Computation

**Franz A. Heinsen**                                                    *franz@glassroom.com*
*GlassRoom Software LLC*

**Leo Kozachkov** *                                                     *leokoz8@gmail.com*
*Thomas J. Watson Research Center, IBM Research*

**Reviewed on OpenReview:** *https://openreview.net/forum?id=SUuzbOSOGu*

## Abstract

Many domains, from deep learning to finance, require compounding real numbers over long sequences, often leading to catastrophic numerical underflow or overflow. We introduce generalized orders of magnitude (GOOMs), a principled extension of traditional orders of magnitude that incorporates floating-point numbers as a special case, and which in practice enables stable computation over significantly larger dynamic ranges of real numbers than previously possible. We implement GOOMs, along with an efficient custom parallel prefix scan, to support native execution on parallel hardware such as GPUs. We demonstrate that our implementation of GOOMs outperforms traditional approaches with three representative experiments, all of which were previously considered impractical or impossible, and now become possible and practical: (1) compounding real matrix products *far* beyond standard floating-point limits; (2) estimating spectra of Lyapunov exponents in parallel, *orders of magnitude faster* than with previous methods, applying a novel selective-resetting method to prevent state colinearity; and (3) capturing long-range dependencies in deep recurrent neural networks with *non-diagonal recurrent states, computed in parallel via a prefix scan, without requiring any form of stabilization.* Our results show that our implementation of GOOMs, combined with efficient parallel scanning, offers a scalable and numerically robust alternative to conventional floating-point numbers for high-dynamic-range applications.[1]

## 1 Introduction

Scientists and engineers often work with real numbers spanning large dynamic ranges, which can exceed the limits of common floating-point formats. A typical example is a chain of real-valued matrix products that fails with catastrophic numerical error because it compounds element values beyond representable bounds. Such chains are ubiquitous in science and engineering. For instance, in deep learning, chains of gradients are multiplied together for backpropagation (LeCun et al., 1988), and whether or not these chains explode or vanish determines training success (Hochreiter & Schmidhuber, 1997; Pascanu et al., 2013; Mikhaeil et al., 2022). In control theory, adjoint equations are used to determine optimal control inputs (Bryson, 2018). In dynamical systems theory, chains of Jacobian matrix values are iteratively applied to estimate Lyapunov exponents (Strogatz, 2018; Pikovsky & Politi, 2016). In numerous fields, including economics and finance, Markov models describe behavior over time with chains of stochastic matrices (Ross, 1995).

Here, we propose generalizing the concept of "order of magnitude" to include the subset of the complex plane that exponentiates elementwise to the real number line, enabling us to represent any real number—positive, zero, or negative—as a complex logarithm that exponentiates to it. We call such complex logarithms "generalized orders of magnitude," or GOOMs for short. As with ordinary orders of magnitude, GOOMs

---

*The author is now at Brown University, Providence, RI.
[1]Source code for replicating our experiments is available at github.com/glassroom/generalized_orders_of_magnitude.

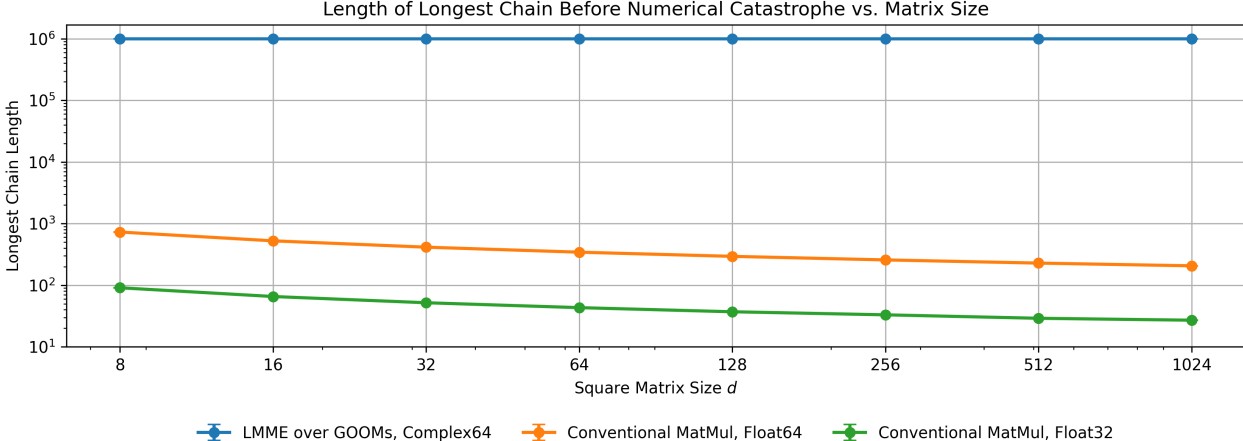

*Figure 1:* Longest chain multiplying random normal square matrices without catastrophic numerical error on an Nvidia GPU, up to a maximum of 1M steps, with real numbers represented as Float32 and Float64, and with GOOMs represented as Complex64, applying a function we call log-matrix-multiplication-exp, or "LMME" (subsection 3.2). Each point represents the mean of 30 runs, with vertical error bars indicating the standard error across those runs. For every run, at each step, we sample matrix elements independently from $\mathcal{N}(0, 1)$.

are more stable than the real numbers to which they exponentiate. Formally, we define GOOMs as a set of mathematical objects, and show that floating-point numbers are a special case of GOOMs. Our work builds upon prior work on logarithmic number systems, with early roots in digital signal processing (Kingsbury & Rayner, 1971; Alsuhli et al., 2023; Kouretas & Paliouras, 2018; Sanyal et al., 2020), though the idea of representing numbers with exponentiation, as in scientific notation, is older. Defining and naming GOOMs enables us to reason about all possible special cases, including floating-point numbers, in the abstract. From a practical standpoint, GOOMs are *complementary* to existing numerical formats, *not a replacement* for them: GOOMs provide a mechanism that can leverage existing numerical formats to enable software applications to operate over a far greater dynamic range of real numbers than previously possible.

We implement GOOMs, as well as common operations over them, including real-valued matrix multiplication, for a widely used software framework, PyTorch (Paszke et al., 2019), and publish our implementation as an open-source software library. Our implementation represents real and imaginary components with either Float32 or Float64 numbers, which, as we mention above, are themselves special cases of GOOMs, in effect forming an "edifice of GOOMs." By design, the dynamic range of our implementation *far* exceeds the dynamic range of Float32 and Float64, as well as the dynamic range of extended floating-point formats, such as Posits (Gustafson & Yonemoto, 2017), proposed as a replacement for conventional ones (in any practical configuration that could replace floating-point formats). We also quantify our implementation's relative error, running time, and memory use compared to Float32 and Float64. More implementations of GOOMs are evidently possible (including, say, implementations that represent real components with a Posit and imaginary ones with a single bit), but we do not explore them here.

We verify that our implementation works as expected with three representative experiments, all of which were previously considered impractical or impossible, and now become possible and practical: First, we compute chains of up to 1M random normal square matrices of size ranging from 8×8 to 1024×1024, with element values that compound to magnitudes *far* beyond the limits representable with Float32 or Float64, but which remain representable as GOOMs with Float32 real and imaginary components—a composite datatype commonly known as "Complex64." When we compute the chains conventionally over Float32 or Float64, they fail early with catastrophic numerical error, as expected. When we compute the chains over Complex64 GOOMs, they complete successfully (Figure 1).

Second, we formulate, implement, and test a parallel algorithm for estimating the spectrum of Lyapunov exponents of dynamical systems, via a prefix scan (Blelloch, 1990), leveraging the greater dynamic range of GOOMs to prevent catastrophic numerical error. We test our parallel algorithm on all dynamical systems

from a dataset spanning multiple scientific disciplines, including astrophysics, climatology, and biochemistry (Gilpin, 2023a;b). On parallel hardware, on all systems, our method computes accurate estimates orders of magnitude faster than previous methods, which are sequential (Figure 3). A key component of our algorithm is a method we devise for conditionally resetting interim states to arbitrary values, as we compute all states in parallel via a prefix scan. We use our selective-resetting method to detect whenever interim deviation states are close to collapsing into colinear vectors in the direction of the eigenvector associated with the largest Lyapunov exponent, and to reset such near-colinear deviation states by replacing them with orthonormal vectors in the same subspace, as we compute all deviation states in parallel, over GOOMs.

Finally, we formulate, implement, and successfully train and test on several tasks a deep recurrent neural network (RNN) whose layers capture sequential dependencies with a non-diagonal state-space model (SSM), computed over GOOMs, executable in parallel via a prefix scan, allowing recurrent state magnitudes to fluctuate freely over time steps, without any form of stabilization. Parallel prefix scans have become more common in deep learning with the introduction of linear SSMs (Gu et al., 2021; 2022; Smith et al., 2023; Gu & Dao, 2023; Yang et al., 2024; Feng et al., 2024; Grazzi et al., 2024). More recent work has extended the use of parallel prefix scans to iteratively compute the trajectories of nonlinear state-space models (Lim et al., 2024; Gonzalez et al., 2024), broadening their potential applicability across more scientific and engineering domains. Our RNN extends the application of parallel prefix scans to non-diagonal SSMs with recurrent states whose elements can fluctuate freely over a greater dynamic range of real values than previously possible, rendering all forms of stabilization unnecessary.

## 2 Generalized Orders of Magnitude

We define the generalized orders of magnitudes, or GOOMs, as the subset of the complex plane, $\mathbb{C}' \in \mathbb{C}$, which exponentiates elementwise to the real number line:

$$\mathbb{C}' \coloneqq \big\{\, x' \in \mathbb{C} \;\big|\; \exp(x') \in \mathbb{R} \,\big\}, \tag{1}$$

treating all elements of $\mathbb{C}'$ whose exponentiated values are equal to each other as GOOM representations of the same real number. For example, $\exp(3 + 2\pi i) = \exp(3 + 4\pi i)$, so we treat $3 + 2\pi i$ and $3 + 4\pi i$ as GOOM representations of the same real number, $\exp(3) \simeq 20.0855$.

Equivalently, we can say that a GOOM's imaginary component must be either a non-zero integer multiple of $\pi i$ or an integer multiple of $2\pi i$, such that the imaginary component's exponentiation is in $\{-1, 1\}$, per Euler's identity. When the imaginary component is an integer multiple of $2\pi i$, the real number to which the GOOM exponentiates is positive; otherwise, the imaginary component is a non-zero integer multiple of $\pi i$, and the real number represented by the GOOM is negative. As a convention, we treat zero in the real number line as non-negative—*i.e.*, we represent it as a positive GOOM. Ordinary orders of magnitude, including log-probabilities, are in the subset of GOOMs with zero imaginary component.

The concept of a complex order of magnitude, defined as $x' = \log x$ for $x \in \mathbb{R}$, may initially appear unconventional; however, it is well-defined since $e^{x'} = x$. We stop short of formally inducing an isomorphism from $\mathbb{R}$ to $\mathbb{C}'$, and vice versa, because we want to keep open the option of applying arbitrary complex-valued transformations that leverage the structure of the complex plane.[2] Our only requirement for a number to be a GOOM in $\mathbb{C}'$ is that it must exponentiate to a number in the real line.

As with ordinary orders of magnitude, GOOMs are more stable than the real numbers to which they exponentiate. We now provide two examples of how GOOMs can be used to perform two standard operations—real scalar multiplication and the dot product of two real-valued vectors—in a way that is inherently more numerically stable than traditional methods.

---

[2]Consider, for example, a deep learning model that processes data in $\mathbb{C}$ and includes a final layer that applies a transformation from $\mathbb{C}$ to $\mathbb{C}'$, thereby allowing the data to be scaled and projected to $\mathbb{R}$. Not coincidentally, our implementation of GOOMs makes instantiating and training such a model straightforward, because it ensures that backpropagation works seamlessly over $\mathbb{C}$, over $\mathbb{C}'$, and across mappings between $\mathbb{C}'$ and $\mathbb{R}$, by extending PyTorch's infrastructure for complex data types.

**Example 1 (Scalar Multiplication in $\mathbb{R}$ Becomes Addition in $\mathbb{C}'$)** *Suppose we are given $d$ real numbers $x_1, x_2, \ldots, x_d$, and we wish to compute their product:*

$$y = x_1 \cdot x_2 \cdots x_d = \prod_{j=1}^{d} x_j.$$

*We can express this product as a* sum *in $\mathbb{C}'$, since, denoting $x'_j = \log x_j$ for $j = 1, 2, \ldots, d$,*

$$\exp(y') = \prod_{j=1}^{d} \exp(x'_j) \qquad \Longrightarrow \qquad y' = \sum_{j=1}^{d} x'_j.$$

**Example 2 (Dot Product in $\mathbb{R}$ Becomes log-sum-exp in $\mathbb{C}'$)** *Suppose we are given two d-dimensional vectors $a$ and $b$ with elements in $\mathbb{R}$. We wish to compute their dot product:*

$$c = a^\top b = \sum_{j=1}^{d} a_j \, b_j.$$

*We can express this dot product as a log-sum-exp (LSE) operation in $\mathbb{C}'$. Denoting $z'$ as the element-wise sum of $a' = \log(a)$ and $b' = \log(b)$,*

$$z'_j := a'_j + b'_j,$$

*we find that $c' = \log(c) = \mathrm{LSE}(z')$. This is because*

$$\exp(z') = \sum_{j=1}^{d} \exp(a'_j) \exp(b'_j) = \sum_{j=1}^{d} \exp(z'_j) \qquad \Longrightarrow \qquad c' = \log c = \log \sum_{j=1}^{d} \exp(z'_j) = \mathrm{LSE}(z').$$

*Note that even if the individual elements of $a$ and $b$ are very large, the GOOM representation in $\mathbb{C}'$ remains numerically stable. For example, suppose $a_j = b_j = \exp(1000)$. Naively computing the dot product in $\mathbb{R}$ fails due to floating-point overflow. However, computing the LSE in $\mathbb{C}'$ works easily, since $z'_j = 2000$, which is well within the capabilities of the numerically stabilized implementation of this operation,* e.g., *in PyTorch.*

## 2.1 Floating-Point Numbers are a Special Case of GOOMs

Given a GOOM $x' = a + bi$, representing a real number $x$, we know from equation 1 that $x = e^{x'} = e^{a+bi} = e^a \times e^{bi} \in \mathbb{R}$. Calling $e$ the "base," $a$ the "exponent," and $e^{bi}$, which is in $\{-1, 1\}$, the "sign," we have:

$$x = \underset{\substack{\uparrow \\ \text{base}}}{\overset{\substack{\text{exponent} \\ \downarrow}}{e^a}} \times \underset{\substack{\uparrow \\ \text{sign}}}{e^{bi}}, \quad x \in \mathbb{R}. \tag{2}$$

If equation 2 seems familiar, it is because floating-point representations of real numbers, which have a base, an exponent, and a sign, are in $\mathbb{C}'$. Their base is 2 instead of $e$, their exponent is encoded in two sequences of bits (one representing a significand or mantissa, the other representing an integer exponent) instead of as the real component of a complex number, and their sign is encoded as a bit instead of as an exponentiated imaginary component, but otherwise they represent real numbers in the same domain. Given a floating-point number $x$ with a signed significand $s$, represented by a fixed number of bits, and integer exponent $n$, also represented by a fixed number of bits, we have:

$$\begin{aligned}
x &= s \times 2^n \\
&= \mathrm{abs}(s) \times 2^n \times \mathrm{sign}(s) \\
&= 2^{\log_2(\mathrm{abs}(s))} \times 2^n \times \mathrm{sign}(s) \\
&= 2^{\log_2(\mathrm{abs}(s))+n} \times \mathrm{sign}(s), \quad x \in \mathbb{R}.
\end{aligned} \tag{3}$$

In other words, *floating-point numbers are a special case of GOOMs.*[3] Unlike floating-point numbers, which are defined in terms of a fixed number of bits, GOOMs are defined more generally in terms of real and imaginary components, independently how those components might be represented in a computer. Also unlike floating-point numbers, GOOMs explicitly leave open the possibility of applying arbitrary transformations that leverage the structure of the complex plane. In practice, GOOMs are *complementary* to existing floating-point formats, providing a mechanism that enables software applications to operate over a greater dynamic range of real numbers than previously possible, without requiring the introduction of and support for new floating-point formats, *e.g.*, in hardware devices like Nvidia GPUs.

## 3 Implementation

We implement GOOMs for PyTorch, a widely used software framework for parallel computation, as Complex64 and Complex128 composite data types, which represent their real and imaginary components as Float32 and Float64 numbers, respectively. That is, we leverage existing floating-point formats, which are special cases of GOOMs (2.1), to represent the real and imaginary components of other GOOMs, for representing real numbers over a greater dynamic range than is possible with existing floating-point formats. In practice, implementing this "edifice of GOOMs" is more straightforward than our description suggests, thanks to the availability of and support for complex data types in PyTorch.

The dynamic range of our implementation is, as expected, *far* greater than that of Float32 and Float64 (Table 1), the two numerical formats with highest dynamic range supported by Nvidia GPUs. It is also *far* greater than that of extended floating-point formats such as Posits in any practical configuration that could replace Float32 or Float64.[4] We also note that our implementation can be readily extended to represent real and imaginary components with Posits or other numerical formats that make different tradeoffs compared to Float32 and Float64, provided such formats are supported by hardware. Other implementations are evidently possible (including, for example, those that represent real components with a Posit and imaginary ones with a single additional bit) but we do not explore them here. It bears repeating that in practice GOOMs are *complementary* to existing numerical formats.

| Representation | Bits | Smallest Normal Magnitude | Largest Normal Magnitude |
|---|---|---|---|
| Float32 | 32 | $10^{-38} \simeq \exp\left(-10^{1.9395}\right)$ | $10^{38} \simeq \exp\left(10^{1.9395}\right)$ |
| Float64 | 64 | $10^{-308} \simeq \exp\left(-10^{2.8506}\right)$ | $10^{308} \simeq \exp\left(10^{2.8506}\right)$ |
| Complex64 GOOM | 64 | $\exp\left(-10^{38}\right)$ | $\exp\left(10^{38}\right)$ |
| Complex128 GOOM | 128 | $\exp\left(-10^{308}\right)$ | $\exp\left(10^{308}\right)$ |

*Table 1:* Dynamic range for Complex64 and Complex128 GOOMs versus Float32 and Float64.

By design, Complex64 and Complex128 GOOMs benefit from the greater precision at smaller magnitudes of Float32 and Float64, respectively, for representing larger magnitudes. Float32 and Float64 represent magnitudes between 0 and 1 with negative exponents, consuming approximately half of all possible exponents in the floating-point format, and magnitudes above 1 with positive exponents, consuming the remaining exponents, such that precision decays as magnitude increases. Complex64 and Complex128 GOOMs thus have greater precision at smaller magnitudes (in fact, the magnitudes become too small to be mapped via exponentiation to Float32 and Float64, respectively), and precision decays below that of the underlying floating-point format as magnitude increases toward the format's maximum representable magnitude, and beyond (Figure 2). We compare the relative error, running time, and memory use of Complex64 and Complex128 GOOMs versus Float32 and Float64 in Appendix D.

---

[3]Extended floating-point numbers, including Posits (Gustafson & Yonemoto, 2017), are a special case too.

[4]For example, the 64-bit configuration of Posits proposed as a drop-in replacement for Float64 by Gustafson & Yonemoto (2017) (Table 3) covers magnitudes from roughly $10^{-299} \simeq \exp(-10^{2.8376})$ to $10^{299} \simeq \exp(10^{2.8376})$, *far* less than our implementation of Complex64 GOOMs, which cover from roughly $\exp(-10^{38})$ to $\exp(10^{38})$, excluding subnormal real components.

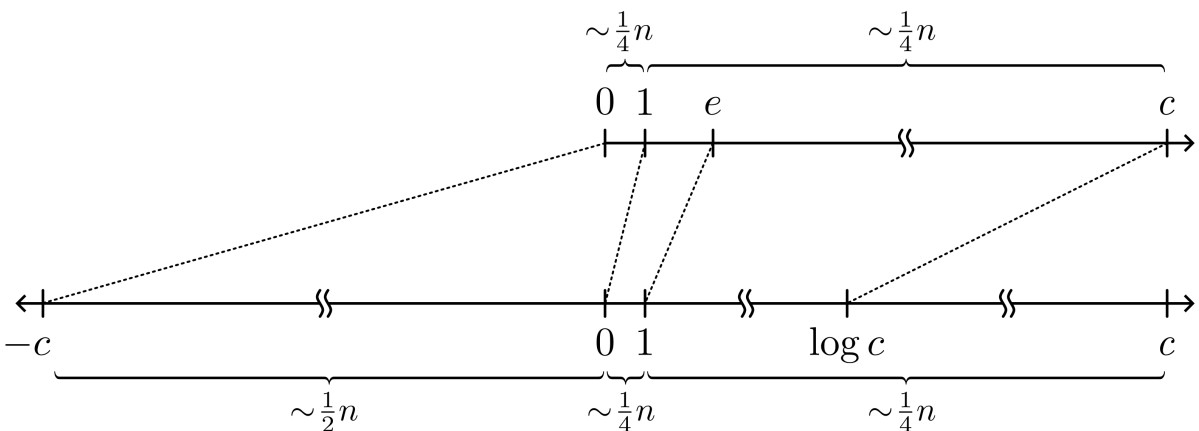

*Figure 2:* At the top, we show the range of magnitudes with positive sign representable by Float32 or Float64, up to a maximum $c$, and their approximate share of $n$, the number of possible bit sequences. For Float32, $n = 2^{32}$; for Float64, $n = 2^{64}$. At the bottom, we show the same magnitudes, mapped to a complex GOOM's real component, represented by the same floating-point format. The shares of $n$ are approximate to account for bitwise differences between Float32 and Float64. For magnitudes with negative sign, the diagram is identical.

Along with functions for mapping existing floating-point numbers to complex-typed GOOMs, and vice versa, we also implement a variety of real-valued functions commonly applied in scientific and engineering applications, including matrix multiplication, over complex-typed GOOMs. As expected, such functions are more stable over $\mathbb{C}'$. For example, scalar product over $\mathbb{R}$ becomes scalar *addition* over $\mathbb{C}'$ (Example 1). A sum over $\mathbb{R}$ becomes a log-sum of exponentials over $\mathbb{C}'$, *i.e.*, it remains a *sum*, but with elementwise nonlinear transformations before and after the sum (Example 2). A sum of products over $\mathbb{R}$, as in a matrix product or tensor contraction, becomes a log-sum-exp of scalar additions over $\mathbb{C}'$, *i.e.*, a *sum of sums*, with elementwise nonlinear transformations before and after the first sum. A composition of linear transformations and elementwise activation functions, as in a deep neural network, becomes a *sum of sums of sums*, with elementwise nonlinear transformations interspersed between sums.

All implemented functions are parallelized, broadcastable over an arbitrary numbers of preceding indices, and compatible with backpropagation of gradients over $\mathbb{C}$, over $\mathbb{C}'$, and over mappings between $\mathbb{C}'$ and $\mathbb{R}$, taking special care to handle the singularity at zero gracefully, for use in a broad range of applications, including deep learning.

### 3.1 Mapping Between Floating-Point Numbers and GOOMs

**Floating-Point Numbers to Complex-Typed GOOMs**   Given a floating-point number $x$, we map it to a complex-typed GOOM $x'$ as follows:

$$x' \longleftarrow \log(x)$$
$$\log(x) := \overset{\text{real}}{\log}\left(\overset{\text{real}}{\text{abs}}(x)\right) + \begin{cases} 0, & x \geq 0 \\ \pi i, & x < 0, \end{cases} \tag{4}$$

where $\overset{\text{real}}{\text{abs}}(\cdot)$ and $\overset{\text{real}}{\log}(\cdot)$ denote custom implementations of real absolute value and elementwise logarithm, respectively, necessary to accommodate a broad range of applications.

Our implementation of $\overset{\text{real}}{\text{abs}}(\cdot)$ redefines its elementwise finite derivative, with respect to its input, to always be non-zero, treating input values equal to zero as non-negative by convention,

$$\frac{d\,\overset{\text{real}}{\text{abs}}(x)}{dx} \;:=\; \begin{cases} 1, & x \geq 0 \\ -1, & x < 0, \end{cases} \qquad \text{// redefined finite derivative} \tag{5}$$

ensuring that gradients are zero in backpropagation only when the backpropagated error is zero. Otherwise, our implementation of real absolute value behaves like PyTorch's default implementation.

Our implementation of $\overset{\text{real}}{\log}(\cdot)$ is configurable, such that for real input elements equal or numerically close to zero, we can specify whether the function will (a) return a sentinel value representing negative infinity, maximizing precision up to the limits of the chosen data type; or (b) return a finite floor value that numerically exponentiates to zero, for applications that require values always to be finite.[5]

We redefine the elementwise finite derivative of $\overset{\text{real}}{\log}(\cdot)$, with respect to its input, to add a data-type-specific small number $\epsilon$ to its denominator,

$$\frac{d\,\overset{\text{real}}{\log}(x)}{dx} \;:=\; \frac{1}{x + \epsilon} \qquad \text{// redefined finite derivative} \tag{6}$$

ensuring that gradients are always finite for backpropagation, including at the singularity for the logarithm of zero. Otherwise, our implementation of real logarithm behaves like PyTorch's default implementation (*e.g.*, it is undefined for negative real input values).

**Complex-Typed GOOMs to Floating-Point Numbers**  Given a complex-typed GOOM $x'$, we map it to a floating-point number $x$ as follows:

$$\begin{aligned} x &\longleftarrow \exp(x') \\ \exp(x') &:= \Re\left( \overset{\text{complex}}{\exp}(x') \right), \end{aligned} \tag{7}$$

where $\overset{\text{complex}}{\exp}(\cdot)$ denotes a custom implementation of complex elementwise exponentiation, and $\Re(\cdot)$ denotes real component. We discard the imaginary component, which may not always be zero due to accumulation of numerical errors. Recall that GOOMs exponentiate to $\mathbb{R}$, by definition.

We redefine the elementwise finite derivative of $\overset{\text{complex}}{\exp}(\cdot)$, with respect to its input, to always be non-zero for backpropagation, by adding to or subtracting a data-type-specific small number $\epsilon$ that shifts the finite derivative's real component away from zero:

$$\frac{d\,\overset{\text{complex}}{\exp}(x')}{dx'} := \overset{\text{complex}}{\exp}(x') + \begin{cases} \epsilon, & \Re\left( \overset{\text{complex}}{\exp}(x') \right) \geq 0 \\ -\epsilon, & \Re\left( \overset{\text{complex}}{\exp}(x') \right) < 0, \end{cases} \qquad \text{// redefined finite derivative} \tag{8}$$

ensuring that gradients are zero in backpropagation only when the backpropagated error is zero. Otherwise, our implementation of complex exponentiation behaves like PyTorch's built-in implementation, relying on PyTorch's default behavior for backpropagating gradients over complex data types.

## 3.2   Real-Valued Matrix Multiplication over Complex-Typed GOOMs

Given two matrices, $A' \in \mathbb{C}'^{n \times d}$ and $B' \in \mathbb{C}'^{d \times m}$, the equivalent of their matrix product over $\mathbb{R}$ is expressible as a log-sum-exp of elementwise additions:

---

[5]In our initial implementation, we specify $\log(\text{SNN}^2)$ as the finite floor value, where SNN is the smallest normal number representable by the chosen floating-point format. For example, for Complex64 GOOMs, which have Float32 components, the smallest normal number representable by Float32 is approximately $1.18 \times 10^{-38}$, so the finite floor, $2 \times \log(1.18 \times 10^{-38})$, is approximately $-174.7$, which numerically exponentiates to zero at Float32 precision.

$$
\begin{aligned}
\mathrm{LMME}\left(A', B'\right) &:= \log\left(\underbrace{\exp(A')\exp(B')}_{\text{MatMul over } \mathbb{R}}\right)\\
&= \log\sum_j \underbrace{\exp(A'_{ij}) \otimes \exp(B'_{jk})}_{\in\, \mathbb{R}^{n\times d\times m}}\\
&= \underset{j}{\mathrm{LSE}}\left(\underbrace{A'_{ij} \oplus B'_{jk}}_{\in\, \mathbb{C}'^{n\times d\times m}}\right),
\end{aligned}
\tag{9}
$$

where LMME is shorthand for "log-matrix-multiplication-exp," LSE denotes log-sum-exp, and $\otimes$ and $\oplus$ denote elementwise product and addition, respectively, of each row of the first matrix with each column of the second one (*i.e.*, $\otimes$ and $\oplus$ denote outer product and addition, respectively, over index $j$). The final expression in equation 9 is easily parallelizable, because each row-column pair can be elementwise summed-then-log-sum-exponentiated independently of all other pairs.

An optimal parallel implementation of LMME requires an efficient parallel implementation of log-sum-exp of pairwise elementwise sums. A straightforward approach to implement it would be to compute all pairwise elementwise sums in parallel, broadcasting over all other dimensions, then apply log-sum-exp, but doing so would be impractical, because it would require $\mathcal{O}(ndm)$ space. Another obvious approach would be to apply log-sum-exp to the elementwise addition of each pair of vectors independently of the other pairs (*e.g.*, with a vector-mapping, or "vmap," operator), but doing so would run into memory-bandwidth constraints on hardware accelerators like Nvidia GPUs, which are better suited for parallelizing computational kernels that execute and aggregate results over tiled sub-tensors. Unfortunately, PyTorch and its ecosystem, including intermediate compilers like Triton (Tillet et al., 2019), currently provide no support for developing highly optimized complex-typed kernels.

As a compromise, we implement $\mathrm{LMME}(A', B')$ so it delegates the bulk of parallel computation to PyTorch's existing, highly optimized, low-level implementation of the dot-product over $\mathbb{R}$, as follows:

$$
\overset{\text{Compromise}}{\mathrm{LMME}}\left(A', B'\right) := \log\left(\underbrace{\exp\left(A'_{ij} - a_i\right)\exp\left(B'_{jk} - b_k\right)}_{\text{Scaled MatMul over } \mathbb{R}}\right) + a_i + b_k,
\tag{10}
$$

where $a_i \in \mathbb{R}^n$ and $b_k \in \mathbb{R}^m$ are vectors with log-scaling constants, necessary because the interim exponentiation to $\mathbb{R}$ may return values outside the bounds representable as floating-point numbers:

$$
\begin{aligned}
a_i &= \overset{\text{elem}}{\max}\left(\max_j\left(\Re\left(A'_{ij}\right)\right),\, 0\right)\\
b_k &= \overset{\text{elem}}{\max}\left(\max_j\left(\Re\left(B'_{jk}\right)\right),\, 0\right),
\end{aligned}
\tag{11}
$$

with $\overset{\text{elem}}{\max}(\cdot)$ denoting elementwise maximum, and $\Re(\cdot)$ denoting elementwise real components. The expression in equation 10 follows from the fact that we can scale each row of the left matrix and each column of the right matrix, before multiplying the two matrices, and subsequently undo the scaling for every element of the resulting product:

$$
\sum_j \exp\left(A'_{ij}\right)\exp\left(B'_{jh}\right) = \left(\sum_j\left(\frac{\exp\left(A'_{ij}\right)}{\exp(a_i)}\right)\left(\frac{\exp\left(B'_{jk}\right)}{\exp(b_k)}\right)\right)\odot\exp(a_i)\odot\exp(b_k),
\tag{12}
$$

where division is applied elementwise, implicitly broadcasting over missing indices, and $\odot$ denotes elementwise (Hadamard) product, also implicitly broadcasting over missing indices. We compute $a_i$ and $b_k$ without impacting gradients in backpropagation (*i.e.*, detached from the computation graph).

We recognize that our initial implementation of LMME is a sub-optimal compromise, both in terms of precision (we execute scaled dot-products over a floating-point format, instead of elementwise sums over complex-typed GOOMs) and performance (we must compute not only a scaled matrix product, but also per-row and per-column maximums on the left and right matrices, respectively, two elementwise subtractions, and two elementwise sums). In practice, we find that our initial implementation of LMME works well in diverse experiments (section 4), incurring execution times that are approximately twice as long as the underlying real-valued matrix product on highly parallel hardware—a reasonable initial tradeoff, in our view, for applications that must be able to handle a greater dynamic range of real magnitudes. See Appendix D for a comparison of our implementation of LMME to PyTorch's matrix product over Float32 and Float64.

### 3.3 Other Real-Valued Functions over Complex-Typed GOOMs

In principle, we can naively formulate the equivalent over $\mathbb{C}'$ of any real-valued function $f$ as a function composition $f'$ that elementwise exponentiates the complex input, applies $f$ to the exponentiated input, and then takes the elementwise logarithm of $f$'s output:

$$f' := \log \circ f \circ \exp. \tag{13}$$

In practice, we can never implement the naive formulation, because the interim exponentiation to $\mathbb{R}$ may return values that fall outside the bounds representable by existing floating-point formats. Instead, we must either (a) implement $f'$ such that it avoids interim exponentiation to $\mathbb{R}$ altogether, always remaining in $\mathbb{C}'$, or (b) scale elements in the log-domain, before exponentiation to $\mathbb{R}$, and undo the scaling after taking the elementwise logarithm, using techniques analogous to the "log-sum-exp trick," but specific to each function $f$. We use both approaches, (a) and (b), to implement over $\mathbb{C}'$ a variety of real-valued functions commonly used in science and engineering.

Please see our published source code for the full list of implemented functions.

## 4 Representative Experiments

We verify that our implementation works as expected with three representative experiments: (a) long chains of real-valued matrix products that compound element magnitudes *far* beyond the limits of Float32 and Float64 numbers; (b) a parallel algorithm for estimating the full spectra *and largest* of Lyapunov exponents of dynamical systems, via a prefix scan, orders of magnitude faster than previous methods, leveraging complex-typed GOOMs to avoid catastrophic numerical error; and (c) a deep RNN whose layers capture sequential dependencies with a non-diagonal linear SSM over complex-typed GOOMs, executable in parallel via a prefix scan, allowing recurrent state magnitudes to fluctuate freely over a greater dynamic range of real numbers than previously possible, making all forms of stabililization unnecessary.

### 4.1 Stability and Dynamic Range of Complex GOOM Matrix Products vs. Conventional Methods

We compare the stability and dynamic range of our implementation of LMME against those of conventional matrix multiplication over $\mathbb{R}$ with floating-point numbers, on long chains of matrix products that compound element magnitudes toward infinity. For $d$ in $\{8, 16, 32, \ldots, 1024\}$, for $t$ in $\{1, 2, 3, \ldots, 10^6\}$, we multiply a chain of random square matrices $A_t \in \mathbb{R}^{d \times d}$, each element of which is independently sampled from a normal distribution $\mathcal{N}(0, 1)$. At each step $t$, the state of a chain is

$$S_t = A_t S_{t-1}, \quad A_t \sim \mathcal{N}(0, 1)^{d \times d}, \tag{14}$$

with $S_0 \sim \mathcal{N}(0, 1)^{d \times d}$ as the initial value. We repeatedly update the chain's state until all steps are completed, or until the computation fails with catastrophic numerical error—whichever occurs first. For each value of $d$, we attempt to complete the chain 30 times on a recent Nvidia GPU.

When we attempt to compute all chains with conventional matrix multiplication over $\mathbb{R}$ with Float32 and Float64 numbers, we find that all chains fail early with catastrophic numerical error.

When we attempt to compute the chains over $\mathbb{C}'$ with our implementation of LMME, representing GOOMs as Complex64 numbers, we find that *all chains successfully complete all steps* (Figure 1).

At each step $t$, the log-state of a chain over $\mathbb{C}'$ is,

$$S'_t = \text{LMME}\left(A'_t, S'_{t-1}\right), \quad A'_t \sim \log \mathcal{N}(0,1)^{d \times d}, \tag{15}$$

with $S'_0 \sim \log \mathcal{N}(0,1)^{d \times d}$ as the initial value. As before, we repeatedly update the chain's state until all steps are completed, or until the computation fails with catastrophic numerical error—whichever occurs first. For each value of $d$, we attempt to complete the chain 30 times on a recent Nvidia GPU.

We repeat all tests on a recent multi-core CPU, and obtain essentially the same results. We do not show plots for the CPU tests because they are indistinguishable from those executed on the GPU. Further analysis reveals that all chain states are representable in their entirety as complex-typed GOOMs, but not as the floating-point numbers to which they would exponentiate, were it possible.

## 4.2 Parallel Estimation of Lyapunov Exponents over Complex-Typed GOOMs

A fundamental concept in the theory of dynamical systems is the Lyapunov exponent (Strogatz, 2018; Pikovsky & Politi, 2016; Bradley & Kantz, 2015; Gilpin, 2024). Loosely speaking, Lyapunov exponents (LEs) quantify how nearby trajectories in the state space of a nonlinear dynamical system diverge (or converge) over time. In general, Lyapunov exponents (LEs) must be computed numerically along specific trajectories of the system under consideration. If the system is stable—for instance, if it is contractive (Lohmiller & Slotine, 1998; Srinivasan & Slotine, 2023)—it may be possible to conclude *a priori* that the LEs are negative. However, assigning precise numerical values to the LEs cannot in general be done analytically. Instead, LEs must be computed by repeatedly multiplying Jacobian matrices of the dynamics. However, this approach can be numerically unstable if the underlying dynamics themselves are unstable. Methods that address this instability, such as iterative QR decomposition (Pikovsky & Politi, 2016), are not parallelizable in time. To understand why, it helps to recall the standard algorithm for computing the largest Lyapunov exponent (cf. (Pikovsky & Politi, 2016, Section 3.1)). The method propagates a vector through a sequence of matrices, normalizing this vector at each step to keep it on the unit sphere. This normalization prevents numerical overflow or underflow. However, because the normalization depends on the current state (specifically, its magnitude), the procedure cannot be implemented using a parallel scan (Blelloch, 1990).

In particular, consider the nonlinear dynamical system

$$x_t \;=\; f_t(x_{t-1}) \;\in\; \mathbb{R}^d, \tag{16}$$

where $f_t$ denotes a given sequence of nonlinear functions. A simple example of equation 16 is a recurrent neural network $x_t = \tanh(W x_{t-1} + b + u_t)$, where $W$ is a weight matrix, $b$ is a bias, and $u_t$ is an input at time $t$. If we wish to understand how small spatial perturbations to solutions of equation 16 influence the future behavior of the system, we may analyze the associated variational equation

$$\delta x_t \;=\; J_t\, \delta x_{t-1}, \qquad \text{where} \qquad J_t := \frac{\partial f_t}{\partial x_{t-1}}. \tag{17}$$

By unrolling the variational equation equation 17 in time, we see that the dynamics of a spatial perturbation to the solution of the nonlinear dynamics equation 16 at time $t = 0$ is entirely determined by the product of Jacobian matrices up to time $t$:

$$\delta x_t \;=\; J_t\, J_{t-1}\, \cdots\, J_1\, \delta x_0 \;=\; \left(\prod_{k=1}^{t} J_k\right)\delta x_0 \;=\; H_t\, \delta x_0,$$

where $H_t$ is defined as the cumulative product of Jacobian matrices up to time $t$. Using $H_t$, the LEs are defined as the long-time average exponential growth (or decay) rates of these infinitesimal perturbations.

Formally, the Lyapunov exponents $\lambda_i$ are given by

$$\lambda_i \;=\; \lim_{t\to\infty} \frac{1}{t} \log \sigma_i\!\left(H_t\right), \tag{18}$$

where $\sigma_i(\cdot)$ denotes the $i$-th singular value of the product of Jacobians and log denotes natural logarithm. In other words, they quantify the average exponential rates at which volumes spanned by perturbation vectors expand or contract under the system's dynamics.

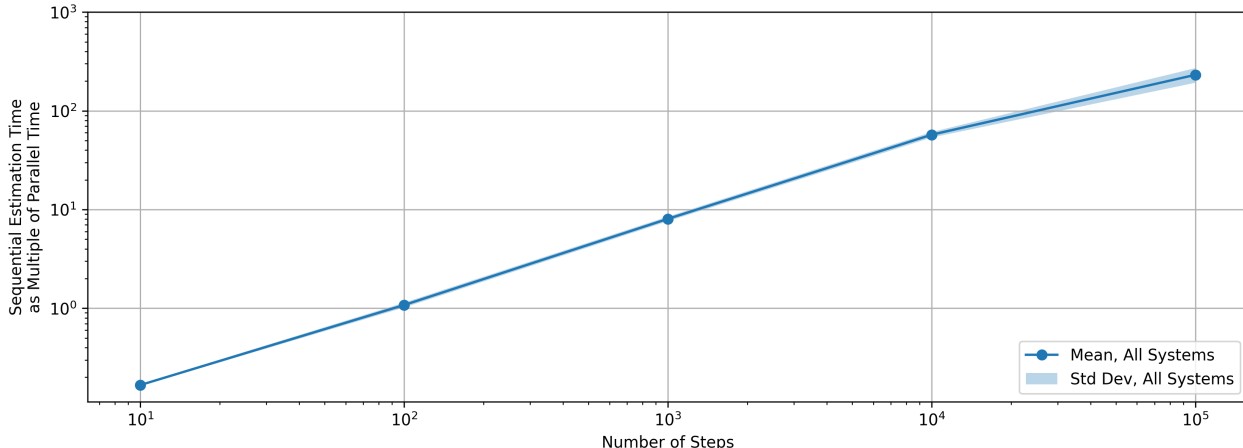

*Figure 3:* Time to estimate the spectrum of Lyapunov exponents sequentially, as a multiple of time to estimate it in parallel, as we increase the number of steps, for all dynamical systems in a dataset spanning multiple scientific disciplines (Gilpin, 2023a;b), in a single Nvidia GPU. The improvement starts tapering off at $10^5$ steps because the GPU's compute capacity is saturated by parallel QR decompositions at all steps. Appendix A shows plots by system.

### 4.2.1 Full Spectra of Lyapunov Exponents in Parallel

The standard approach for estimating the full spectra of Lyapunov exponents, $\Lambda = \begin{bmatrix} \lambda_1 \\ \lambda_2 \\ \vdots \\ \lambda_d \end{bmatrix}$, is:

$$\overset{\text{est}}{\Lambda} := \frac{1}{\Delta t T} \sum_{t=1}^{T} \log\left(\text{abs}\left(\text{Diag}\left((R_t)\right)\right)\right), \tag{19}$$

where

$$\begin{aligned} Q_t, R_t &= \text{QR}(S_t) && \text{// QR-decomposition} \\ S_t &= J_t Q_{t-1}, \end{aligned} \tag{20}$$

given initial deviation states with unit norms $S_0 \in \mathbb{R}^{d \times d}$. That is, at each step $t$ we obtain a triangular matrix $R_t$ by a QR-decomposition of $S_t$, obtained by applying $J_t$ to the preceding orthonormal basis $Q_{t-1}$ obtained by a previous QR-decomposition. The elementwise logarithms of $R_t$'s eigenvalues, in its diagonal, are unscaled estimates of $\Lambda$ at step $t$; their scaled mean is the estimate of $\Lambda$.

Parallel estimation of $\Lambda$ requires not only that we handle magnitudes that may not be representable as floating-point numbers, which we already know we can do with complex-typed GOOMs, but also that we execute a QR-decomposition *before* and *after* the application of each Jacobian matrix $J_t$, to obtain its input state's orthonormal basis $Q_{t-1}$ and its output state's triangular factor $R_t$, respectively. Alas, if we

naively attempt to compute all deviation state matrices in parallel, via a prefix scan, with the intention of subsequently executing all QR decompositions in parallel, we are not able to do so, because states tend to collapse—*i.e.*, become colinear—in the direction of the eigenvector associated with the largest Lyapunov exponent. We must find a way to prevent vectors from collapsing into colinear states.

Our solution is an algorithm that can compute all deviation state matrices in parallel, by applying a method we devise for conditionally resetting interim states to arbitrary values in any linear recurrence, as we compute it in parallel via a prefix scan. Our conditional-resetting method, which we describe separately in section 5, enables us to detect whenever any interim deviation states are close to collapsing into colinear vectors, and to reset such near-colinear states by replacing them with orthonormal vectors in the same subspace, at any time step, as we compute all deviation states in parallel, via a prefix scan, over complex-typed GOOMs. We describe our parallel algorithm below.

**Parallel Algorithm**   We execute the following groups of parallelized computations sequentially:

(a) Compute input states, $S_0, S_1, S_2, \ldots, S_{T-1}$, by cumulatively applying all Jacobian matrices except the last one, $J_1, J_2, \ldots, J_{T-1}$, to the given initial state $S_0$, in parallel, via a prefix scan, over complex-typed GOOMs (to be able to handle all magnitudes), applying our selective-resetting method (section 5) to reset any interim states in the chain that are close to becoming colinear, replacing them with an orthonormal basis obtained via QR-decomposition, which we apply in parallel only to those interim states, if any, selected for resetting. We define "close to becoming colinear" as the cosine similarity of any pair of state vectors exceeding a specified threshold. Before applying QR-decomposition to any near-colinear interim states, we first log-scale them to log-unit norms, over complex-typed GOOMs, and then exponentiate them to values representable as floating-point numbers.

(b) Compute orthonormal input bases $Q_0, Q_1, Q_2, \ldots, Q_{T-1}$, by applying QR-decomposition to every $S_t$. Before applying each such QR-decomposition, we first log-scale its input vectors to log-unit norms, over complex-typed GOOMs, and then exponentiate them to values representable as floating-point numbers. We execute every QR-decomposition independently of the others, in parallel.

(c) Compute output states, $S_1^*, S_2^*, \ldots, S_T^*$, by applying each Jacobian matrix $J_t$ to its preceding input basis $Q_{t-1}$. We apply each Jacobian matrix to its input basis independently of the others, in parallel.

(d) Compute the estimated $\Lambda$ by applying QR-decomposition to every output state $S_t^*$, extracting the diagonal elements (eigenvalues) of each triangular matrix $R_t$, taking the elementwise logarithm of their absolute values, and computing their mean over all states. We apply QR-decomposition and extract and transform diagonal elements for each state independently of the others, in parallel.

With $T$-way parallelism, fixing $d$, our algorithm's time complexity is $\mathcal{O}(\log T)$, because the parallel prefix scan in group (a) above has $\mathcal{O}(\log T)$ time complexity, and every subsequent group of computations has $\mathcal{O}(1)$ time complexity, except for the final mean computed in group (d), which has $\mathcal{O}(\log T)$ time complexity. We test our parallel algorithm on all dynamical systems from a dataset spanning multiple disciplines, including astrophysics, climatology, and biochemistry (Gilpin, 2023a;b), on a recent Nvidia GPU. We find that our parallel algorithm computes accurate estimates, orders of magnitude faster than sequential estimation with previous methods (Figure 3).

### 4.2.2   Largest Lyapunov Exponent in Parallel

Estimating only the largest Lyapunov exponent (LLE) in parallel is straightforward over complex-typed GOOMs, thanks to their ability to handle a greater dynamic range of magnitudes. The standard approach for estimating the LLE relies on measuring changes in norm of an initial deviation vector $u_0 \in \mathbb{R}^d$ with unit norm:

$$\overset{\text{est}}{\text{LLE}} := \frac{1}{\Delta t T} \sum_{t=1}^{T} \log \left( \frac{\|s_t\|}{\|u_{t-1}\|} \right), \tag{21}$$

where $\Delta t$ is the discrete time interval, and

$$
\begin{aligned}
s_t &= J_t u_{t-1} \\
u_t &= \frac{s_t}{\|s_t\|}.
\end{aligned}
\tag{22}
$$

Normalization of each preceding deviation state is necessary to keep element magnitudes from compounding above or below the bounds representable by existing floating-point formats. Without normalization of preceding steps, equation 21 becomes:

$$
\overset{\text{est}}{\text{LLE}} := \frac{1}{\Delta t T} \sum_{t=1}^{T} \log\left(\frac{\|s_t\|}{\|s_{t-1}\|}\right), \quad s_t = J_t s_{t-1}, \quad s_0 = u_0.
\tag{23}
$$

In Appendix B, we show that equation 23 simplifies to:

$$
\overset{\text{est}}{\text{LLE}} := \frac{1}{2\Delta t T} \text{LSE}(2\,\text{PSCAN}(\text{LMME})(J_T', \ldots, J_2', J_1', u_0'))
\tag{24}
$$

where $\text{PSCAN}(\text{LMME})(\cdot)$ denotes a parallel prefix scan applying LMME over time steps, and

$$
\text{For each } z \in \{u_0, J_1, J_2, \ldots, J_T\}, \quad z' \longleftarrow \log(z).
$$

We test our method for parallel LLE estimation on all dynamical systems from a dataset spanning multiple disciplines, including astrophysics, climatology, and biochemistry (Gilpin, 2023a;b), on a recent Nvidia GPU. We find that our parallel method computes accurate estimates, and as expected, its execution is orders of magnitude faster than sequential estimation with previous methods.

### 4.3 Parallelizable Non-diagonal State-Space RNN over Complex-Typed GOOMs

Deep learning models apply a variety of normalization methods (RMSNorm, LayerNorm, BatchNorm, Group-Norm, etc.), residual mechanisms (LSTM cells, GRU cells, residual layers, skip connections, etc.), gated linear activation functions (ReLU, SiLU, GeLU, GLU, SwiGLU, etc.), and various other techniques to keep the magnitudes of feedforward elements and backpropagated gradients from becoming too small or too large, for preventing catastrophic numerical errors. Deep RNN models tend to be most susceptible to such numerical issues, because their layers capture sequential dependencies by repeatedly applying the same recurrent transformation at each time step, compounding the transformation.

We formulate and implement a deep RNN whose layers capture sequential dependencies with non-diagonal recurrences over complex-typed GOOMs, allowing recurrent state magnitudes to fluctuate freely over a greater dynamic range of real numbers than previously possible, *enabling computation in parallel via a prefix scan without requiring any form of stabilization*. Otherwise, the RNN operates conventionally over floating-point numbers. The RNN consists of a standard embedding layer that maps token values (*e.g.*, integers representing symbols in a vocabulary) to feature vectors representing token states; multiple residual recurrent layers that capture sequential dependencies over complex-typed GOOMs, and a conventional model head that is task-specific (*e.g.*, a linear transformation for classification tasks).

Each residual recurrent layer captures sequential dependencies on multiple heads per token, by applying: first, LayerNorm and a linear transformation with bias to obtain each token's heads; second, a parallel prefix scan of a non-diagonal linear recurrence, capturing per-head sequential dependencies with a standard state-space model, (over complex-typed GOOMs, as we will detail shortly); and finally, gated linear units (GLU) on every head, followed by a linear transformation of the flattened heads, to obtain a residual, added to the token's input state.

Per head, we capture sequential dependencies with a standard linear state-space model:

$$x_t = Ax_{t-1} + Bu_t$$
$$y_t = Cx_t + Du_t,$$

(25)

where $u_t \in \mathbb{R}^d$, $x_t \in \mathbb{R}^d$, and $y_t \in \mathbb{R}^h$, with $h = 2d$ for application of GLU, and $A$, $B$, $C$, and $D$ are matrix parameters—but we compute the recurrence *over complex-typed GOOMs*. That is, we first map the initial state $x_0$, all input head states $u_t$, and all matrix parameters to complex-typed GOOMs,

$$\text{For each } z \in \{x_0, u_t, A, B, C, D\}, \quad z' \longleftarrow \log(z),$$

then compute the recurrent relationship in equation 25 over complex-typed GOOMs, for every head, in parallel, via a prefix scan that obtains all states $x_t'$ *without applying any form of stabilization*,

$$x_t' = \text{LSE}\left(\text{LMME}(A', x_{t-1}'), \text{LMME}(B', u_t')\right),$$

(26)

then map $x_t'$ back to a floating-point format, for executing the remaining computations in the layer. We cannot exponentiate the elements of $x_t'$ outright, because their magnitudes might not be representable by a floating-point format, so we first log-scale $x_t'$, then exponentiate its elements, as follows:

$$c \longleftarrow \max\left(\Re(x_t')\right)$$
$$\text{scaled } x_t \longleftarrow \exp\left(x_t' - c + 2\right),$$

(27)

computing the log-scaling real constant $c$ without altering gradients in backpropagation (*i.e.*, detached from the computation graph). The elements of every scaled $x_t$ fall between $-\exp(2)$ and $\exp(2)$.

We train and test instances of our RNN on several tasks, including (but not limited to) generative language modeling on The Pile (Gao et al., 2020), classification and generation of pixel sequences from MNIST (LeCun et al., 2010), and a Copy Memory task. Perhaps the most remarkable finding about the training dynamics is how unremarkable they are, even though we are computing *non-diagonal recurrences in parallel without any form of stabilization*. We are even able to compile and autocast to Float16 all components of the RNN that execute operations over floating-point formats, executing in PyTorch's eager mode only the non-diagonal recurrences in parallel, via the prefix scan over complex-typed GOOMs. Figure 4 shows examples of training runs for two tasks. Please see our published source code to replicate training runs on all tasks.

## 5 Selective-Resetting Method for Parallel Scans of Linear Recurrences

As part of our work to formulate an algorithm for parallel estimation of Lyapunov exponents (subsection 4.2), we devise a method for conditionally resetting interim states to arbitrary values in a linear recurrence, as we compute all of its states in parallel, via a prefix scan. For ease of exposition, we describe our method with a non-diagonal, time-variant, linear recurrence over $\mathbb{R}$, understanding that the method generalizes to *any linear recurrence*—diagonal or not, time-variant or not—over $\mathbb{R}$ or other fields.

Given:

- an initial state $X_0$ in $\mathbb{R}^{d \times d}$,

- a sequence of transition matrices $A_1, A_2, \ldots, A_n$, each in $\mathbb{R}^{d \times d}$,

- a "selection" function $\mathcal{S} : \mathbb{R}^{d \times d} \mapsto \{0, 1\}$, and

- a "reset" function $\mathcal{R} : \mathbb{R}^{d \times d} \mapsto \mathbb{R}^{d \times d}$,

we wish to compute a linear recurrence $X_t = A_t X_{t-1}$, for $t$ in $\{1, 2, \ldots, n\}$, via a parallel prefix scan, but modifying the recurrence, such that, for any interim compound state $A^*$ computed during the parallel scan,

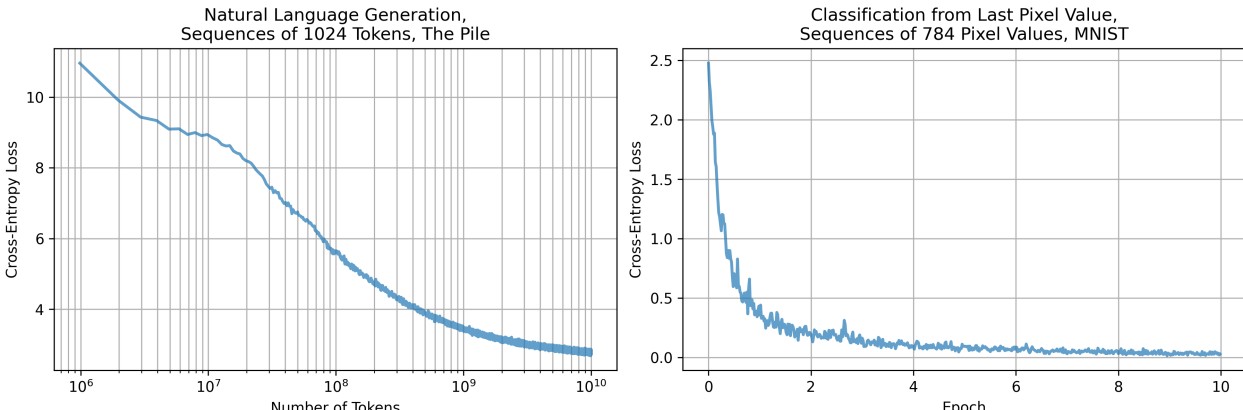

*Figure 4:* Examples of training dynamics for the RNN we implement, capturing sequential dependencies with non-diagonal recurrences, computed in parallel via a prefix scan, *without any form of stabilization.* Left: Natural language generation on The Pile (Gao et al., 2020), with a 124M-parameter RNN incorporating a 50257 token-id vocabulary and 24 layers; we stopped training at 10B tokens. Right: Classification, from last pixel value, of sequences of 784 pixels from MNIST (LeCun et al., 2010), with a 12.8M-parameter RNN incorporating a 256 token-id vocabulary and 8 layers. See our source code for replicating all training runs, including for both tasks shown here.

if $\mathcal{S}(A^*) = 1$, we reset $A^*$ by replacing it with $\mathcal{R}(A^*)$, making $\mathcal{R}(A^*)$ the new initial state for subsequent states, until we reach either (a) the final state of the recurrence, or (b) a subsequent state that has previously been selectively reset—whichever occurs first.

We accomplish our objective in two steps. First, we add a bias matrix at each step: $X_t = A_t X_{t-1} + B_t$, with every bias matrix initialized to all-zeros: $B_t \longleftarrow \{0\}^{d \times d}$, for $t$ in $\{1, 2, \ldots, n\}$, such that, at initialization, the new recurrence matches the original unmodified one.

Second, we execute a parallel prefix scan over the new recurrence, but *instead of applying an ordinary linear recurrence with bias*, we apply the following binary associative transformation to every pair of tuples containing preceding interim states, $\left(A^*_{\text{prev}}, B^*_{\text{prev}}\right)$, and subsequent ones to be updated, $\left(A^*_{\text{curr}}, B^*_{\text{curr}}\right)$:

$$
\begin{aligned}
&\text{// first, selective resetting:} \\
&\text{If } \mathcal{S}\left(A^*_{\text{prev}}\right) = 1 \text{ and } B^*_{\text{prev}} = \{0\}^{d \times d}: \\
&\qquad B^*_{\text{prev}} \longleftarrow \mathcal{R}\left(A^*_{\text{prev}}\right) \\
&\qquad A^*_{\text{prev}} \longleftarrow \{0\}^{d \times d} \\[4pt]
&\text{// then, ordinary recurrence:} \\
&A^*_{\text{curr}} \longleftarrow A^*_{\text{curr}} A^*_{\text{prev}} \\
&B^*_{\text{curr}} \longleftarrow A^*_{\text{curr}} B^*_{\text{prev}} + B^*_{\text{curr}}
\end{aligned}
\tag{28}
$$

We obtain a modified sequence of states that may or may not match the original sequence, because one or more of its states may have been reset during the scan. The selective-resetting transformation equation 28 is associative, because it can reset each state only once, and the zeroed-out transition matrix of any reset state eventually zeroes-out all subsequent transition matrices via cumulative multiplication.

Our selective-resetting method incorporates multiple moving parts, and it may take some effort to grasp fully how all of them interact upon a first read. For the convenience of readers seeking an intuitive understanding of the method, Appendix C explains it more informally with step-by-step examples.

## 6   Discussion

By encoding real numbers as complex logarithms, GOOMs enable stable, scalable and parallel computation across dynamic ranges far beyond what Float32 or Float64 support, without requiring changes to hardware or floating-point standards.

Our experiments support GOOMs' robustness and versatility. In long matrix product chains, complex-typed GOOMs avoid overflow and underflow where conventional approaches fail early. In Lyapunov exponent estimation, the combination of complex-typed GOOMs with a parallel prefix scan and a novel selective-resetting method enables accurate, time-parallel analysis of chaotic systems, orders of magnitude faster than with previous methods. In deep learning, RNNs handle freely fluctuating non-diagonal recurrent states over complex-typed GOOMs without compounding beyond representable limits or degrading gradients, rendering all forms of stabilization unnecessary.

We show that GOOMs can be implemented straightforwardly in native PyTorch, supporting autograd and GPU execution through standard complex types. We note that although the current implementation of operations like log-matrix-multiplication-exp (Section 3.2) introduces some overhead, future work on custom kernels could close this gap.

To summarize, GOOMs in practice offer a robust, software-level solution to numerical instability in high-dynamic-range computations. Their blend of flexibility, precision, scalability, and parallelizability makes them a powerful tool for scientific computing in general, and deep learning in particular.

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

## A    Time to Estimate Spectrum of Lyapunov Exponents by System

The plots below show time to estimate the spectrum of Lyapunov exponents sequentially, as a multiple of time to estimate it in parallel with our algorithm, as we increase the number of time steps, for every dynamical system in a dataset spanning multiple scientific disciplines (Gilpin, 2023a;b). Each point is the mean of seven runs on a single Nvidia GPU. In a few cases (most notably, "MacArthur"), the improvement in execution time tapers off as we increase the number of steps to and above $10^5$, because parallel computation of all QR decompositions saturates the single GPU at approximately 100% utilization.

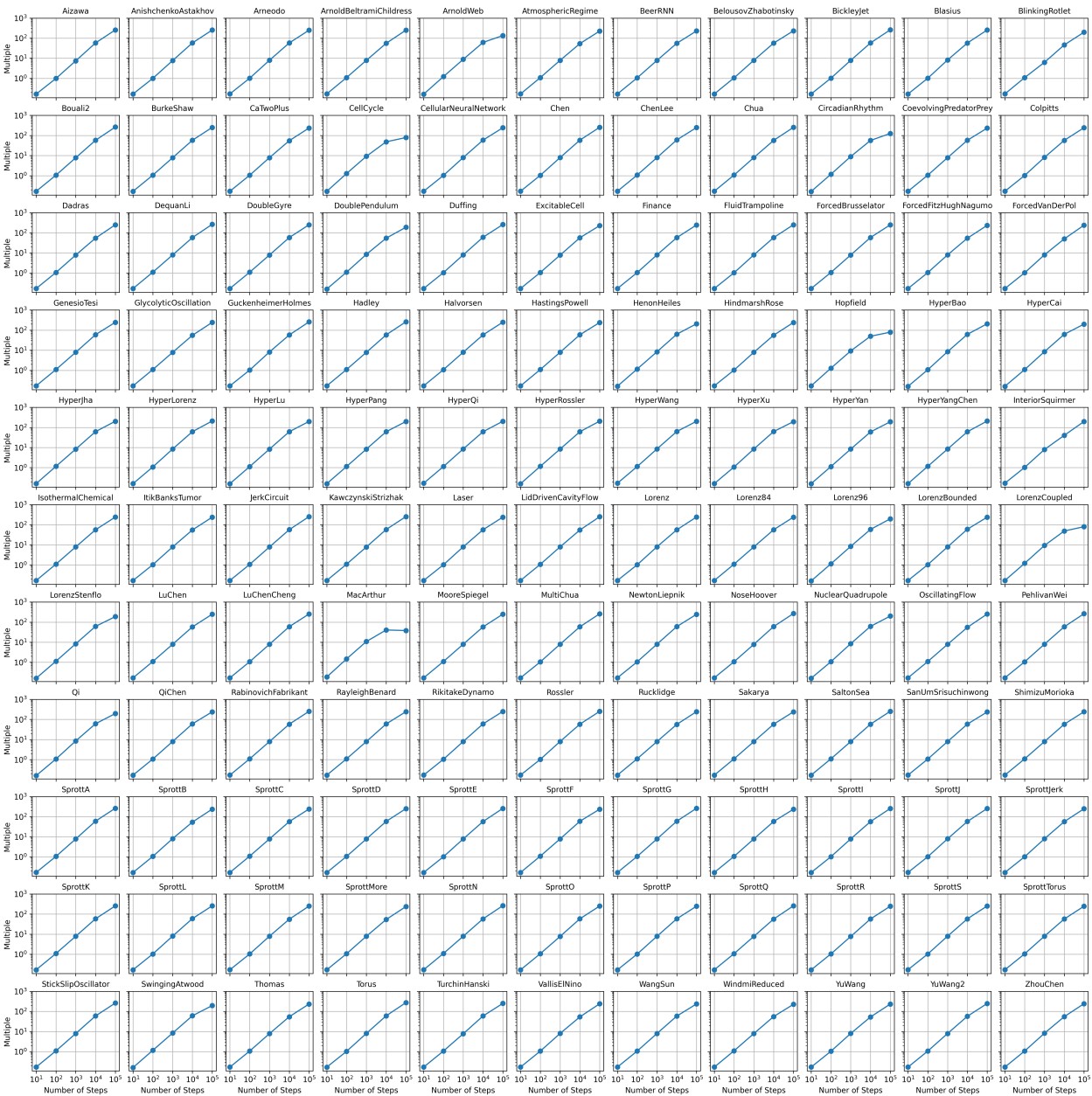

## B   Derivation of Expression for Estimating Largest Lyapunov Exponent

If we express the sum of scalar logarithms in equation 23 as a logarithm of scalar products, we obtain:

$$
\begin{aligned}
\overset{\text{est}}{\text{LLE}} &= \frac{1}{\Delta t T} \log \prod_{t=1}^{T} \left( \frac{\|s_t\|}{\|s_{t-1}\|} \right), \quad s_0 = u_0 \\
&= \frac{1}{\Delta t T} \log \left( \frac{\|s_T\|}{\|s_{T-1}\|} \times \cdots \times \frac{\|s_2\|}{\|s_1\|} \times \frac{\|s_1\|}{\|u_0\|} \right) \\
&= \frac{1}{\Delta t T} \log \frac{\|s_T\|}{\|u_0\|} \\
&= \frac{1}{\Delta t T} \log \| J_T \ldots J_2 J_1 u_0 \| \\
&= \frac{1}{\Delta t T} \log \underbrace{\sqrt{\sum (J_T \ldots J_2 J_1 u_0)^2}}_{\text{Norm computed over } \mathbb{R}} \\
&= \frac{1}{2\Delta t T} \log \sum (J_T \ldots J_2 J_1 u_0)^2 \\
&= \frac{1}{2\Delta t T} \log \sum \exp(\log(J_T \ldots J_2 J_1 u_0))^2 \\
&= \frac{1}{2\Delta t T} \text{LSE} \left( 2 \, \text{LMME}(\log J_T, \text{LMME}(\ldots, \text{LMME}(\log J_2, \text{LMME}(\log J_1, \log u_0)))) \right) \\
&= \frac{1}{2\Delta t T} \text{LSE} \left( 2 \, \text{LMME}(J_T', \text{LMME}(\ldots, \text{LMME}(J_2', \text{LMME}(J_1', u_0')))) \right) \\
&= \frac{1}{2\Delta t T} \text{LSE}(2 \, \text{PSCAN}(\text{LMME})(J_T', \ldots, J_2', J_1', u_0')).
\end{aligned}
\tag{29}
$$

## C Intuition behind Selective Resetting, with Step-by-Step Examples

Consider, for a moment, the following time-variant non-diagonal linear recurrence with biases,

$$X_t = A_t X_{t-1} + B_t, \tag{30}$$

where $X_t$, $A_t$, $B_t$, and initial state $X_0$ are all square matrices in $\mathbb{R}^{d \times d}$, for $t$ in $\{1, 2, \ldots, n\}$. If we zero-out a particular transition matrix $A_t$, we "reset" the recurrence at that step, with the corresponding $B_t$ becoming the new initial state. For example, given a recurrence with $n = 3$ states,

$$
\begin{aligned}
X_1 &= A_1 X_0 + B_1 \quad \text{// } X_0 \text{ is the initial state} \\
X_2 &= A_2 X_1 + B_2 \\
X_3 &= A_3 X_2 + B_3,
\end{aligned}
\tag{31}
$$

if we set $A_2 \longleftarrow \{0\}^{d \times d}$, then $B_2$ becomes the new initial state for subsequent states:

$$
\begin{aligned}
X_1 &= A_1 X_0 + B_1 \\
X_2 &= \cancel{A_2 X_1} + B_2 \quad \text{// } B_2 \text{ becomes the initial state for subsequent states} \\
X_3 &= A_3 X_2 + B_3.
\end{aligned}
\tag{32}
$$

We can apply the same mechanism to reset linear recurrences *without biases*. Initialize all $B_t$'s to all-zeros, such that, at initialization, we have the equivalent of a recurrence without biases:

$$X_t = A_t X_{t-1} + \cancel{B_t}. \quad \text{// all } B_t \text{'s are zeroed-out at initialization} \tag{33}$$

Now, at any step, we can reset this linear recurrence without biases, by zeroing-out that step's transition matrix and replacing the corresponding bias matrix with a new initial state of our choice. For example, a sequence of $n = 3$ states with all $B_t$'s initialized to all-zeros, is:

$$
\begin{aligned}
X_1 &= A_1 X_0 + \cancel{B_1} & &= A_1 X_0 \quad \text{// } X_0 \text{ is the initial state} \\
X_2 &= A_2 X_1 + \cancel{B_2} & &= A_2 X_1 \\
X_3 &= A_3 X_2 + \cancel{B_3} & &= A_3 X_2.
\end{aligned}
\tag{34}
$$

If we set, say, $B_2 \longleftarrow \mathcal{R}(A_2 X_1)$, then $A_2 \longleftarrow \{0\}^{d \times d}$, where $\mathcal{R} : \mathbb{R}^{d \times d} \mapsto \mathbb{R}^{d \times d}$ is a function of our choice, we obtain the following sequence of states:

$$
\begin{aligned}
X_1 &= A_1 X_0 + \cancel{B_1} & &= A_1 X_0 \\
X_2 &= \cancel{A_2 X_1} + B_2 & &= B_2 \quad \text{// } B_2 \text{ becomes the new initial state for all subsequent states} \\
X_3 &= A_3 X_2 + \cancel{B_3} & &= A_3 X_2.
\end{aligned}
\tag{35}
$$

Now, consider a parallel prefix scan, applied right-to-left, over the unmodified recurrence.

### C.1 Parallel Prefix Scan without Selective Resets

When we apply a parallel prefix scan, right-to-left, over the unmodified recurrence, we have:

Parallel Prefix Scan:

$$
\begin{array}{ll}
\text{Input states :} & \boxed{\begin{array}{c} A_3 \\ B_3 \end{array}} \quad \boxed{\begin{array}{c} A_2 \\ B_2 \end{array}} \quad \boxed{\begin{array}{c} A_1 \\ B_1 \end{array}} \quad \boxed{\begin{array}{c} X_0 \\ B_0 \end{array}} \quad \text{\scriptsize // bias states initialized to all-zeros} \\[2em]

\text{Parallel step 1 :} & \boxed{\begin{array}{c} A_3 A_2 \\ A_3 B_2 + B_3 \end{array}} \overset{\longleftarrow}{\scriptstyle\text{update}} \boxed{\begin{array}{c} A_2 \\ B_2 \end{array}} \quad \boxed{\begin{array}{c} A_1 X_0 \\ A_1 B_0 + B_1 \end{array}} \overset{\longleftarrow}{\scriptstyle\text{update}} \boxed{\begin{array}{c} X_0 \\ B_0 \end{array}} \\[2em]

\text{Parallel step 2 :} & \boxed{\begin{array}{c|c} A_3 A_2 (A_1 X_0) & A_2 (A_1 X_0) \\ A_3 A_2 (A_1 B_0 + B_1) + A_3 B_2 + B_3 & A_2 (A_1 B_0 + B_1) + B_2 \end{array}} \overset{\longleftarrow}{\scriptstyle\text{update}} \boxed{\begin{array}{c|c} A_1 X_0 & X_0 \\ A_1 B_0 + B_1 & B_0 \end{array}} \\[2em]

\text{Output states :} & \boxed{\begin{array}{c|c|c|c} A_3 A_2 A_1 X_0 & A_2 A_1 X_0 & A_1 X_0 & X_0 \\ \{0\}^{d \times d} & \{0\}^{d \times d} & \{0\}^{d \times d} & \{0\}^{d \times d} \end{array}} \quad \text{\scriptsize // $B_t$'s are all-zeros} \\[2em]

\text{Sums :} & \boxed{\begin{array}{c|c|c|c} A_3 A_2 A_1 X_0 & A_2 A_1 X_0 & A_1 X_0 & X_0 \end{array}} . \quad \text{\scriptsize // same as recurrence without biases}
\end{array}
\tag{36}
$$

That is, we obtain $X_t = A_t X_{t-1}$, equal to the original recurrence without biases.

### C.2 Parallel Prefix Scan with One Selective Reset

If we selectively reset, say, the second interim compound state, before parallel step 2, we have:

Parallel Prefix Scan, *Selectively Reset before Parallel Step 2*:

$$
\begin{array}{ll}
\text{Input states :} & \boxed{\begin{array}{c} A_3 \\ B_3 \end{array}} \quad \boxed{\begin{array}{c} A_2 \\ B_2 \end{array}} \quad \boxed{\begin{array}{c} A_1 \\ B_1 \end{array}} \quad \boxed{\begin{array}{c} X_0 \\ B_0 \end{array}} \quad \text{\scriptsize // bias states initialized to all-zeros} \\[2em]

\text{Parallel step 1 :} & \boxed{\begin{array}{c} A_3 A_2 \\ A_3 B_2 + B_3 \end{array}} \overset{\longleftarrow}{\scriptstyle\text{update}} \boxed{\begin{array}{c} A_2 \\ B_2 \end{array}} \quad \boxed{\begin{array}{c} A_1 X_0 \\ A_1 B_0 + B_1 \end{array}} \overset{\longleftarrow}{\scriptstyle\text{update}} \boxed{\begin{array}{c} X_0 \\ B_0 \end{array}} \\[2em]

\textit{Selective reset :} & \text{Replace } \boxed{\begin{array}{c} A_1 X_0 \\ A_1 B_0 + B_1 \end{array}} \text{ with } \boxed{\begin{array}{c} \{0\}^{d \times d} \\ \mathcal{R}(A_1 X_0) \end{array}} \\[2em]

\text{Parallel step 2 :} & \boxed{\begin{array}{c|c} A_3 A_2 (\{0\}^{d \times d}) & A_2 (\{0\}^{d \times d}) \\ A_3 A_2 \, \mathcal{R}(A_1 X_0) + A_3 B_2 + B_3 & A_2 \, \mathcal{R}(A_1 X_0) + B_2 \end{array}} \overset{\longleftarrow}{\scriptstyle\text{update}} \boxed{\begin{array}{c|c} \{0\}^{d \times d} & X_0 \\ \mathcal{R}(A_1 X_0) & B_0 \end{array}} \\[2em]

\text{Output states :} & \boxed{\begin{array}{c|c|c|c} \{0\}^{d \times d} & \{0\}^{d \times d} & \{0\}^{d \times d} & X_0 \\ A_3 A_2 \, \mathcal{R}(A_1 X_0) & A_2 \, \mathcal{R}(A_1 X_0) & \mathcal{R}(A_1 X_0) & \{0\}^{d \times d} \end{array}} \quad \text{\scriptsize // $B_t$'s are all-zeros} \\[2em]

\text{Sums :} & \boxed{\begin{array}{c|c|c|c} A_3 A_2 \, \mathcal{R}(A_1 X_0) & A_2 \, \mathcal{R}(A_1 X_0) & \mathcal{R}(A_1 X_0) & X_0 \end{array}} . \quad \text{\scriptsize // modified linear recurrence}
\end{array}
\tag{37}
$$

That is, we obtain a modified sequence in which step 2's state was selectively reset, changing its value from $A_1 X_0$ to $\mathcal{R}(A_1 X_0)$, compounded in all subsequent states, as we computed all states in parallel.

# D   Quantitative Comparisons of Our Implementation to Floating-Point Formats

We compare Complex128 and Complex64 GOOMs, respectively, to Float64 and Float32, the two float formats with greatest precision and dynamic range currently supported on Nvidia GPUs. The comparisons, shown in the pages that follow, are valid only for our implementation, not for GOOMs in general.

**Magnitude of Errors**   We compare the magnitude of errors for one- and two-argument scalar functions that can be used to compose many other functions, and also for a representative matrix product. The one-argument scalar functions are: reciprocal $y = 1/x$, square root $y = \sqrt{x}$, square $y = x^2$, natural logarithm $y = \log x$, and exponential $y = e^x$. The two-argument scalar functions are: addition $z = x + y$ (and implicitly, subtraction, since $x - y = x + -y$) and scalar product $z = xy$ (and implicitly, division, since $x/y = xy^{-1}$). The matrix product is of two $1024 \times 1024$ matrices with elements independently sampled from $\mathcal{N}(0, 1)$, representative of typical matrix products in deep learning models. We apply all one- and two-argument functions except the exponential function over a range of values spanning the approximate number of decimal digits to which each float format is precise: $10^{-15}$ to $10^{15}$ for Float64, and $10^{-6}$ to $10^6$ for Float32. We apply the exponential function over a range of values spanning $10^{-5}$ to 10. For one-argument functions, the range consists of 1M input values, equally spaced in the domain of decimal digits. For two-argument functions, each argument's range consists of 10,000 input values, equally spaced in the domain of decimal digits, such that we apply each two-argument function to 100M different pairs of input values. For all one- and two-argument functions, we measure the magnitude of absolute errors in base 10 (i.e., the number of decimal digits of error) in relation to the same operation executed over Float128, a float format with greater precision and dynamic range than Float64 and Float32. For the matrix product, we measure the error normalized by the Frobenius norm of the matrix product in relation to the same operation executed over Float128. We execute all operations on a recent Nvidia GPU, except for operations and comparisons over Float128, because Nvidia GPUs currently do not support it. We implement all transformations over complex-typed GOOMs so they accept *floats* as inputs, internally map them to complex-typed GOOMs, execute all computations over complex-typed GOOMs, map them back to *floats*, and, finally, return those floats as outputs. *Mapping complex-typed GOOMs to floats impacts precision but is necessary for comparison to Float128.* We find that the magnitude of errors varies from approximately the same to only slightly more, notwithstanding the impact of mapping complex-typed GOOMs to floats.

**Running Time**   We compare running time for every one- and two-argument scalar function, as we apply it to a batch of 100M input samples, each independently drawn from $\mathcal{U}(0, 1)$, processing each batch in parallel on a recent Nvidia GPU. We also compare running time for the representative matrix product. We apply each transformation 30 times, each time to a different batch, and compute the mean running time over all batches. We report the mean running time of each transformation over Complex128 and Complex64 GOOMs as a multiple of the mean running time for Float64 and Float32, respectively. We implement every transformation over complex-typed GOOMs to accept complex-typed GOOMs as inputs and return complex-typed GOOMs as outputs, regardless of whether doing so penalizes running time. For example, we implement scalar addition over complex-typed GOOMs as $\log(e^{x'} + e^{y'})$, not as $e^{x'} + e^{y'}$, significantly impacting running time due to the creation of float tensors for storing interim exponentiated values and complex tensors for storing output values. On the other hand, our implementation of natural logarithm incurs zero running time, because complex-typed GOOMs are already natural logarithms. We find that for most operations, the running time of our implementation is approximately twice that of floats.

**Memory Use**   We compare memory use for every one- and two-argument scalar function, as we apply it to a batch of 100M input samples, each independently drawn from $\mathcal{U}(0, 1)$, processing each batch in parallel on a recent Nvidia GPU. We also compare memory use for the representative matrix product. For both complex-typed GOOMs and floats, we measure peak memory allocated, including creation of input, interim, and output tensors, as well as PyTorch overhead, if any. We report peak memory allocated over Complex128 and Complex64 GOOMs as a multiple of peak memory allocated over Float64 and Float32, respectively. We find that peak memory allocated is typically at least twice that of floats, but sometimes it can be less.

For all comparisons, see the pages that follow. To replicate the comparisons, see our published source code.

### D.1 Complex128 GOOMs versus Float64

We show the comparisons of Complex128 GOOMs to Float64 first, because they are likely more important to researchers and practitioners interested in quantitative comparisons of precision.

### D.1.1 Magnitude of Errors on Reciprocals

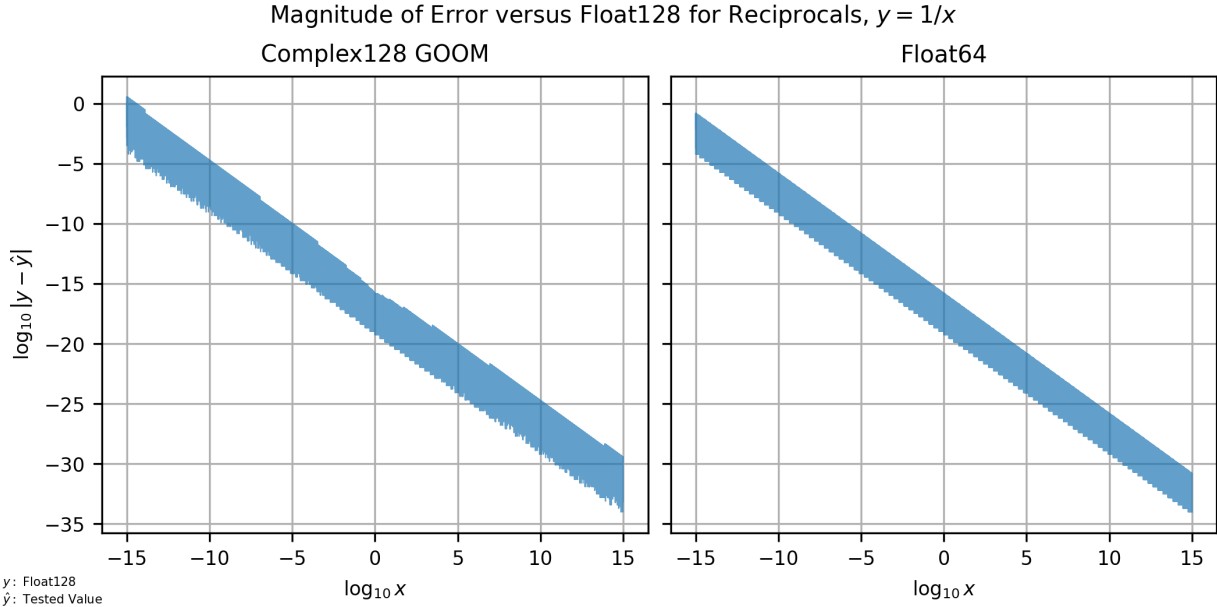

### D.1.2 Magnitude of Errors on Square Roots

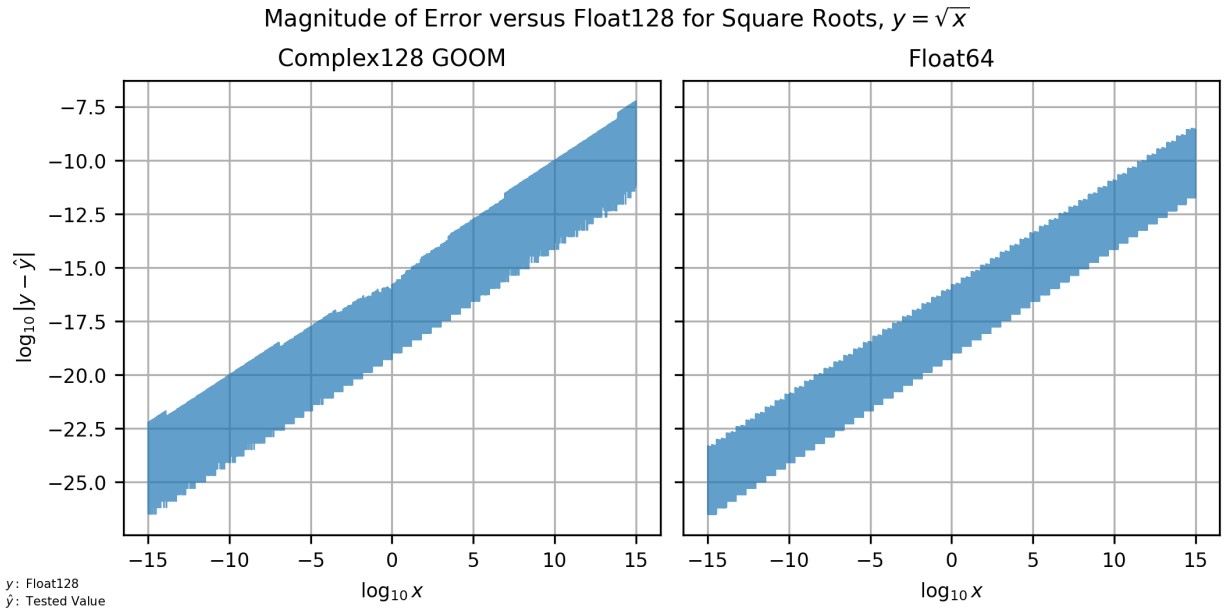

### D.1.3 Magnitude of Errors on Squares

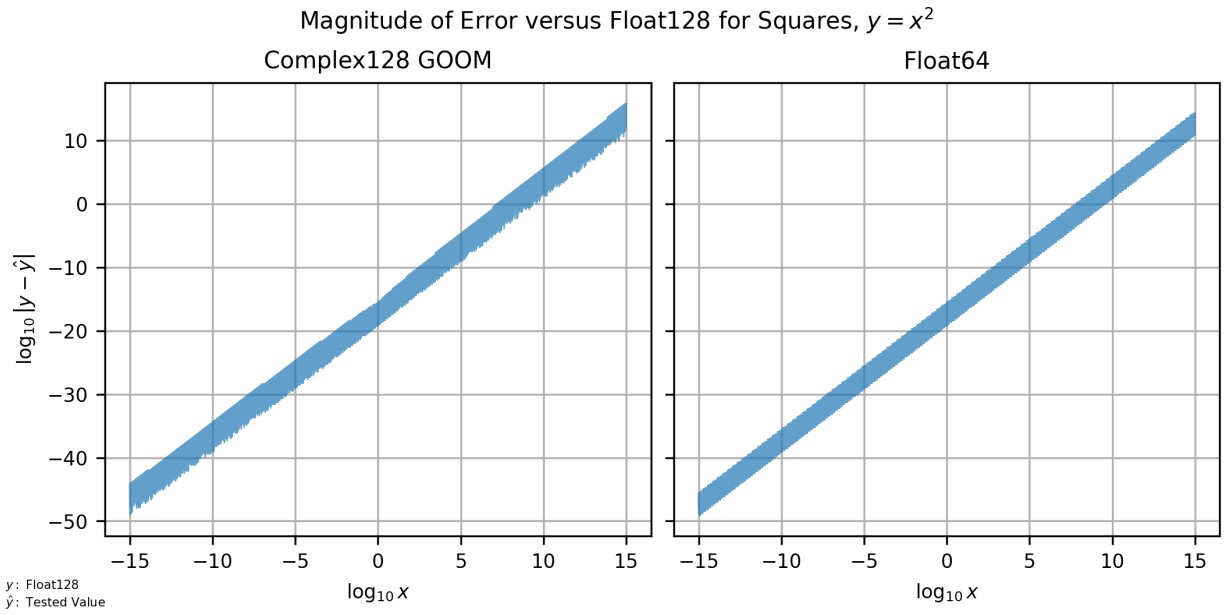

### D.1.4 Magnitude of Errors on Natural Logarithms

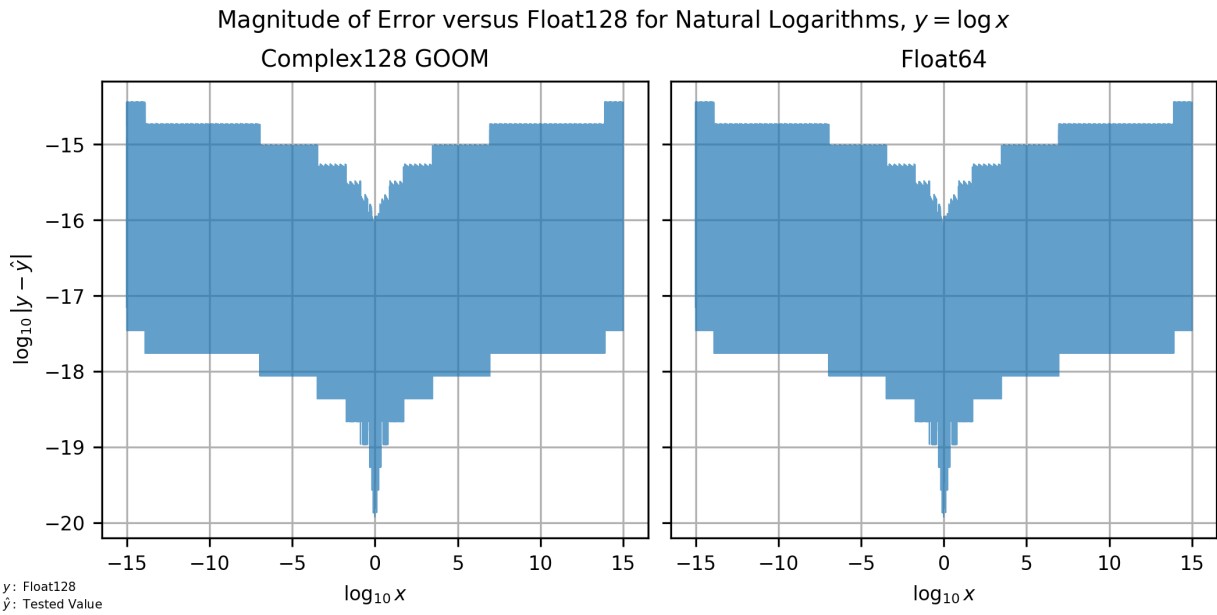

### D.1.5 Magnitude of Errors on Exponentials

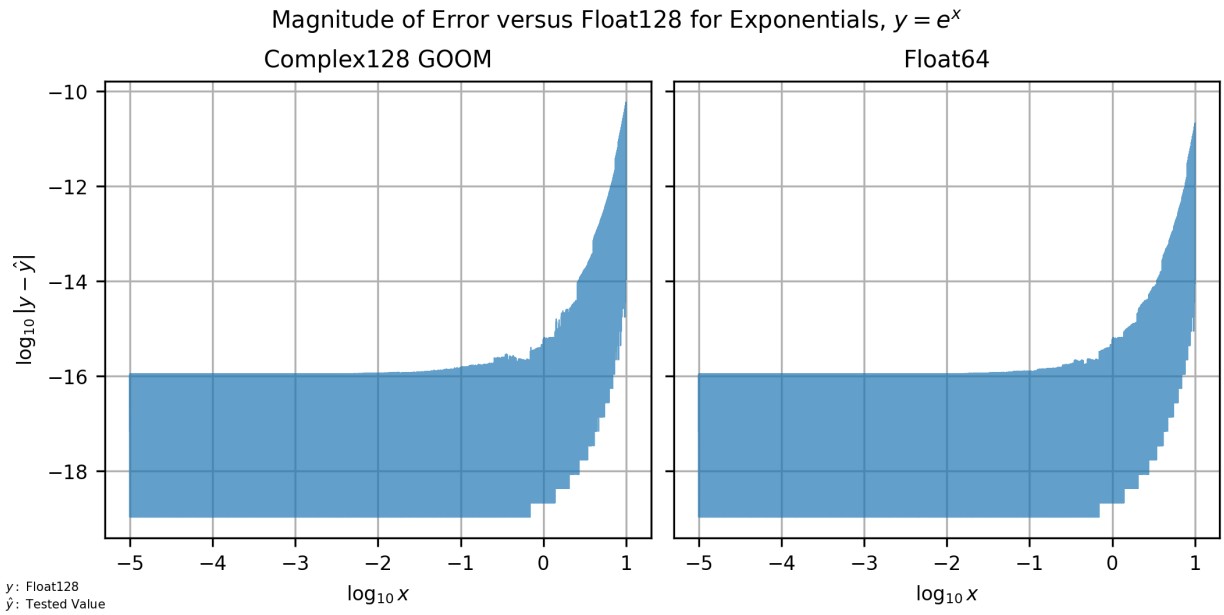

### D.1.6 Magnitude of Errors on Scalar Addition/Subtraction

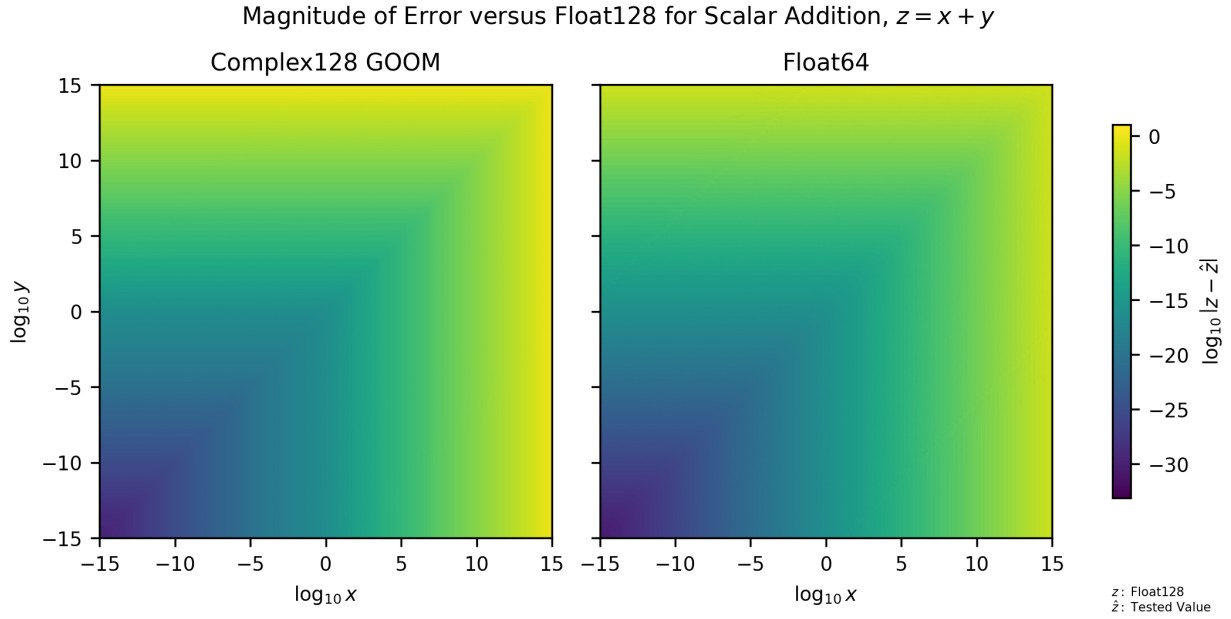

### D.1.7  Magnitude of Errors on Scalar Product/Division

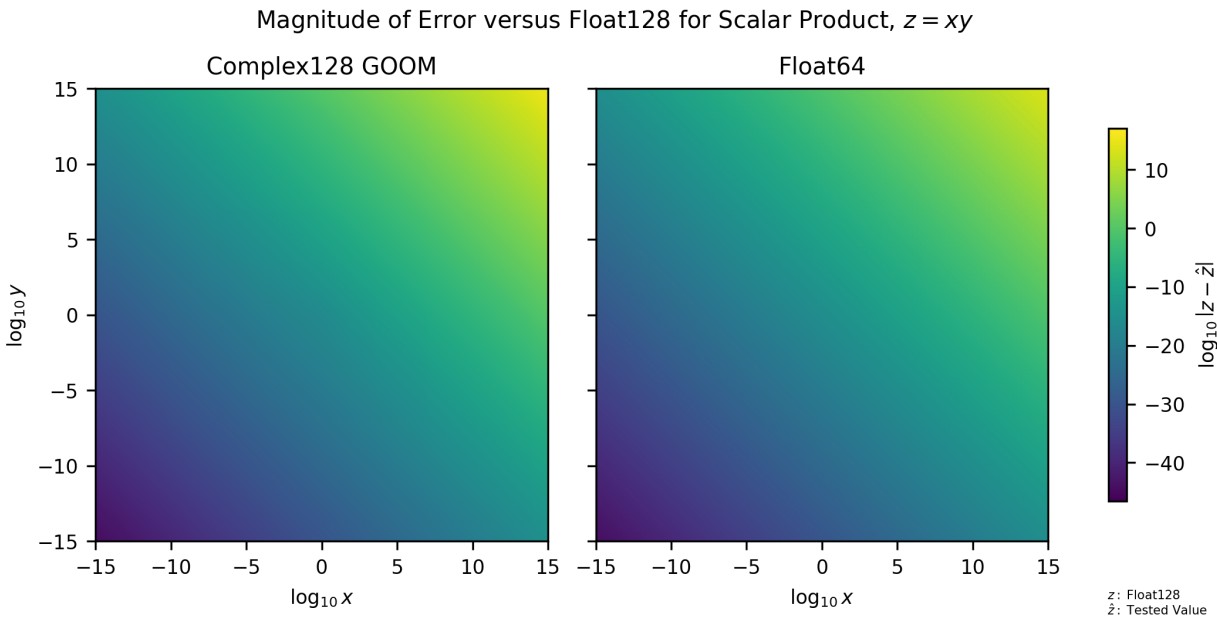

### D.1.8  Normalized Errors on Representative Matrix Product

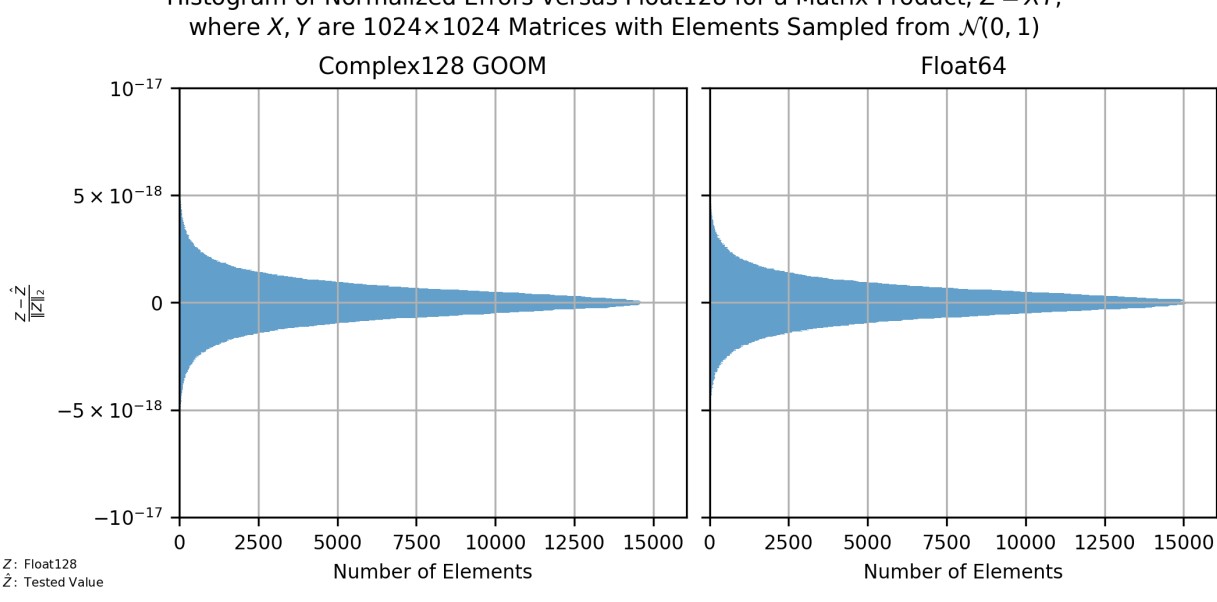

### D.1.9 Execution Times

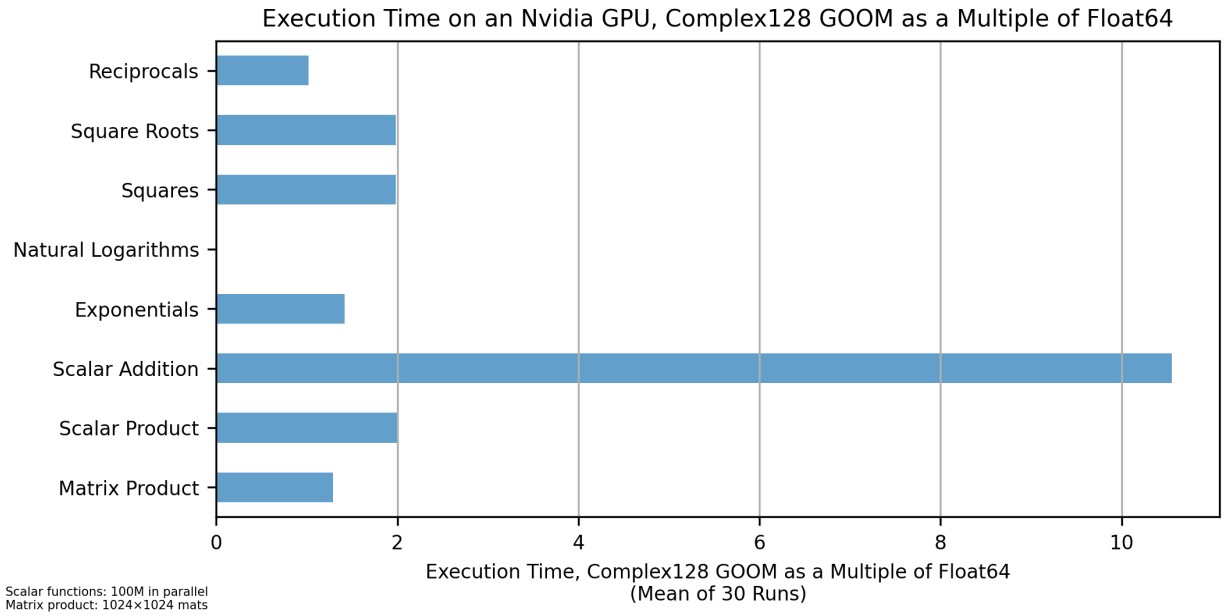

Mean execution times are as reported by `torch.utils.benchmark.Timer`. To the extent possible, we apply transformations in-place, to minimize the impact of new memory allocations on execution time. Complex128 GOOMs are already natural logarithms, so obtaining them incurs no computation.

### D.1.10 Peak Memory Allocated

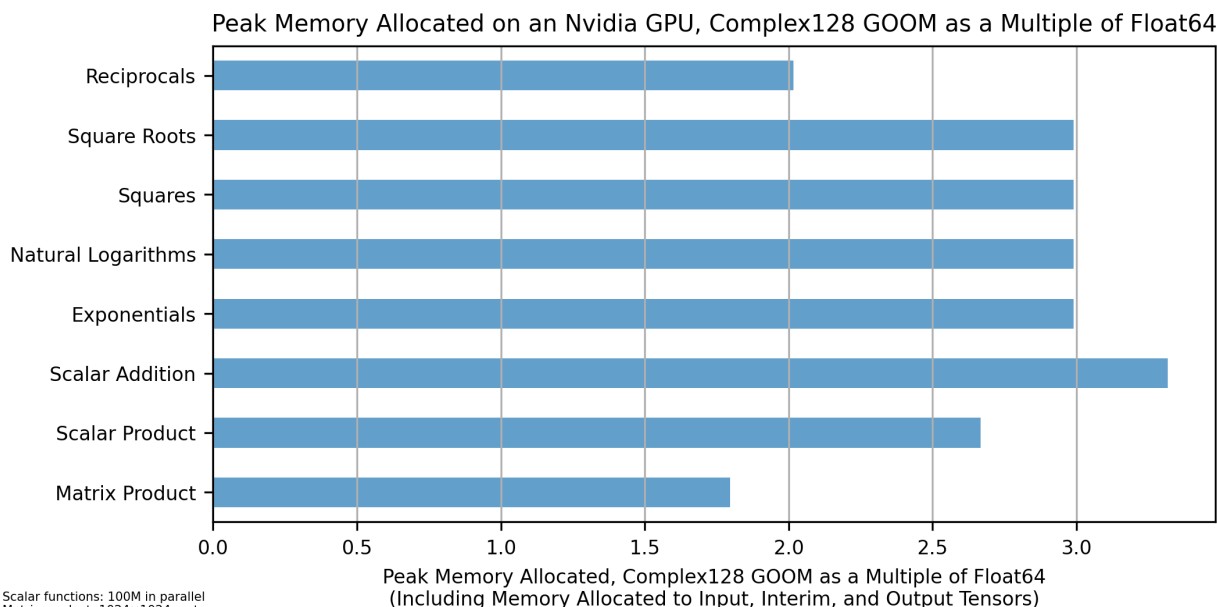

Memory allocation figures are as reported by `torch.cuda.memory.max_memory_allocated`. To the extent possible, we do *not* apply transformations in-place, to allocate memory for input, interim, and output tensors. Memory use by the matrix product over Complex128 GOOMs is for our initial implementation of LMME.

### D.2 Complex64 GOOMs versus Float32

We show the comparisons of Complex64 GOOMs to Float32 in the same order as the comparisons of Complex128 GOOMs to Float64, for easier cross-reference.

### D.2.1 Magnitude of Errors on Reciprocals

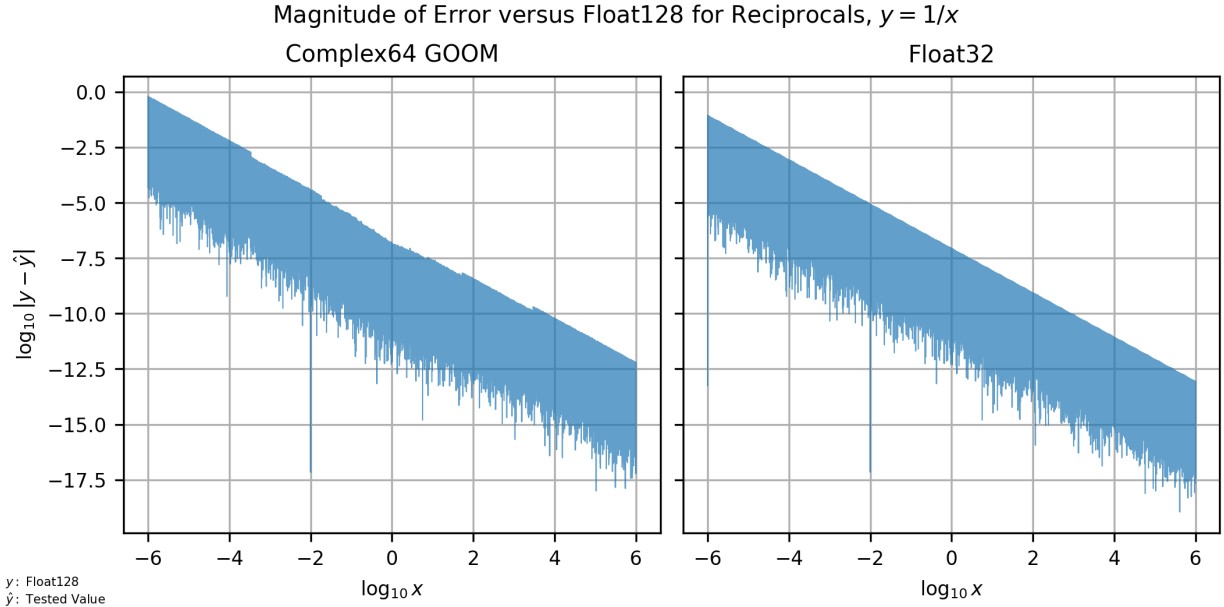

### D.2.2 Magnitude of Errors on Square Roots

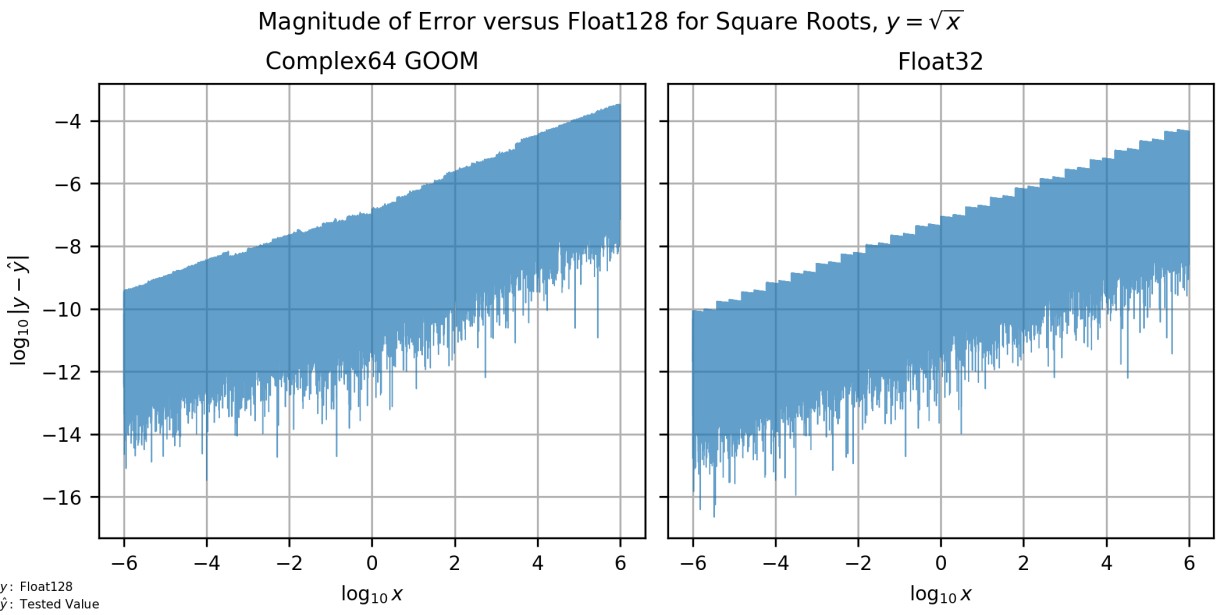

### D.2.3 Magnitude of Errors on Squares

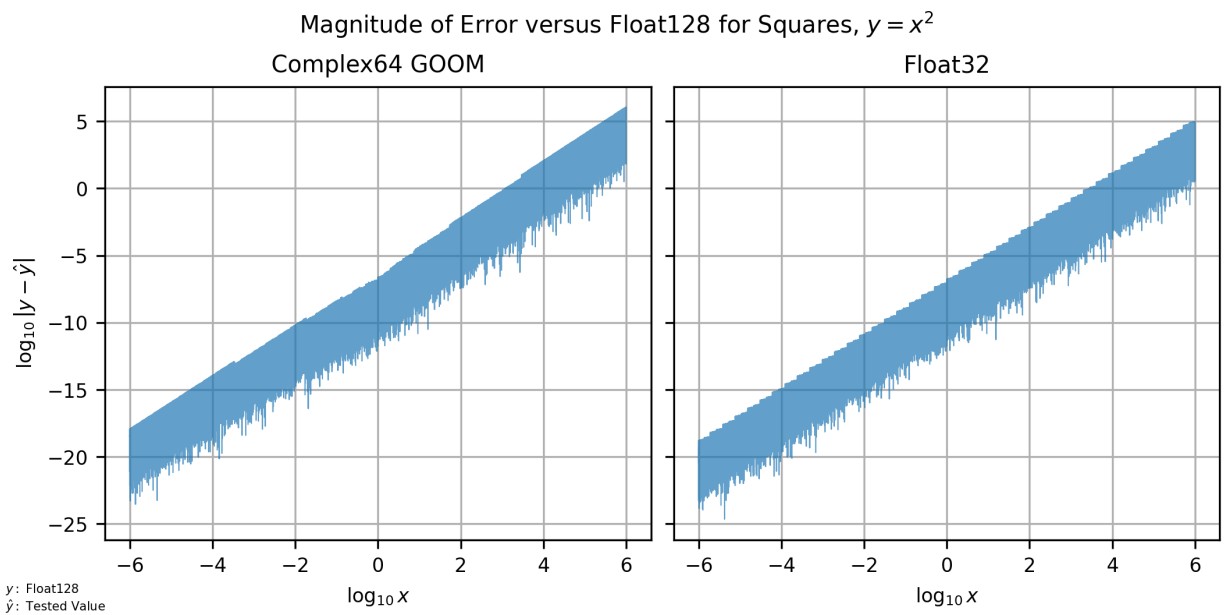

### D.2.4 Magnitude of Errors on Natural Logarithms

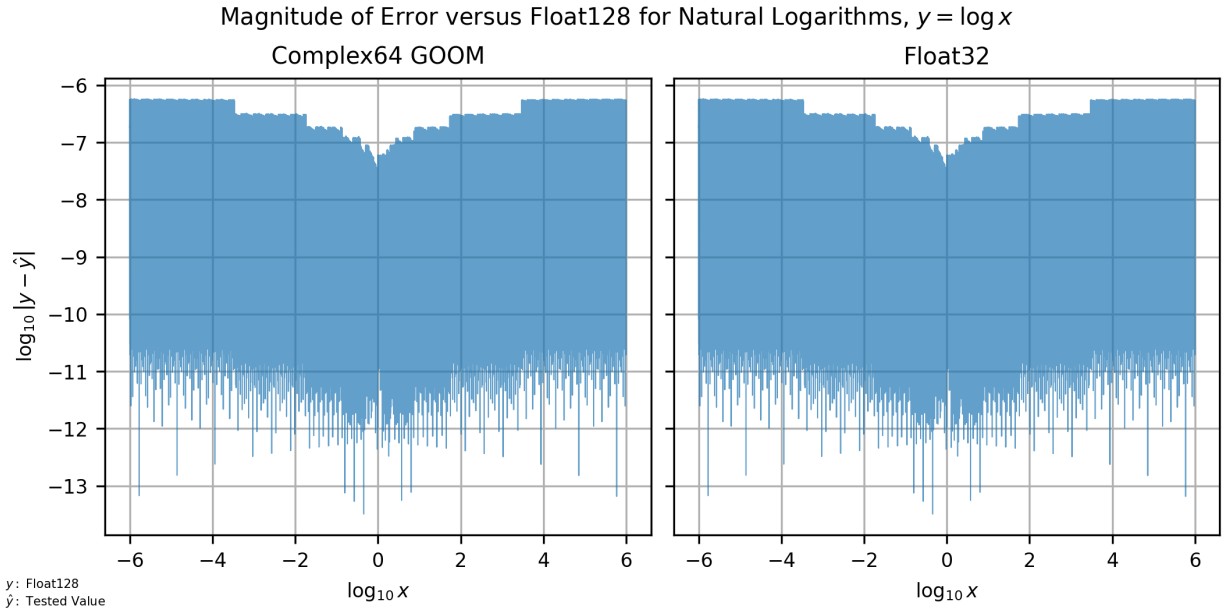

### D.2.5 Magnitude of Errors on Exponentials

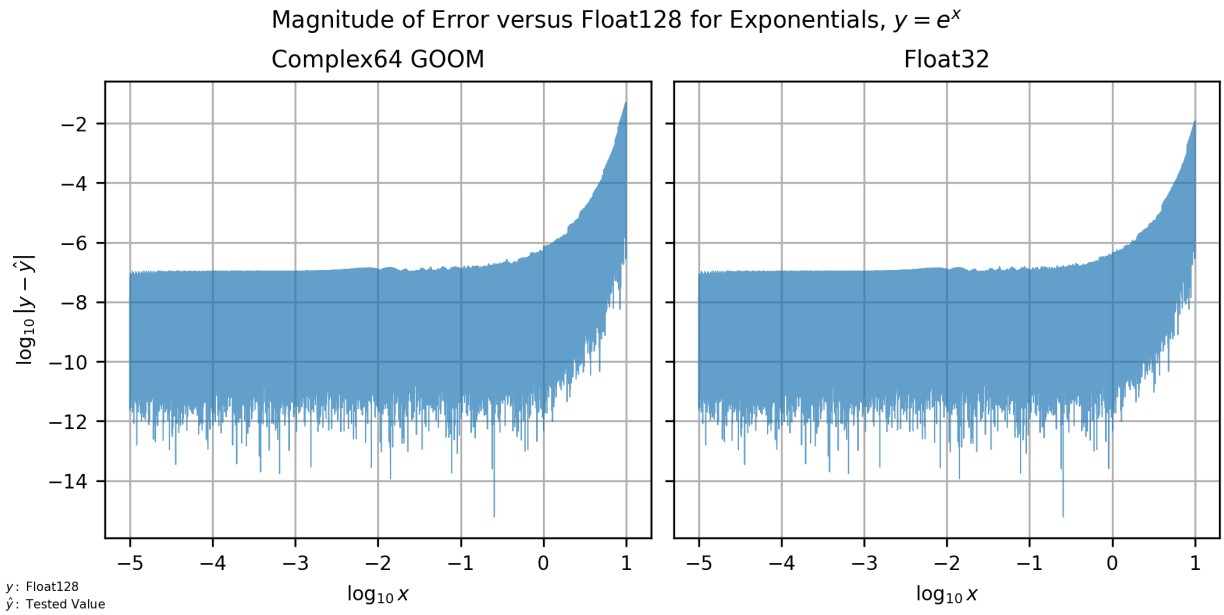

### D.2.6 Magnitude of Errors on Scalar Addition/Subtraction

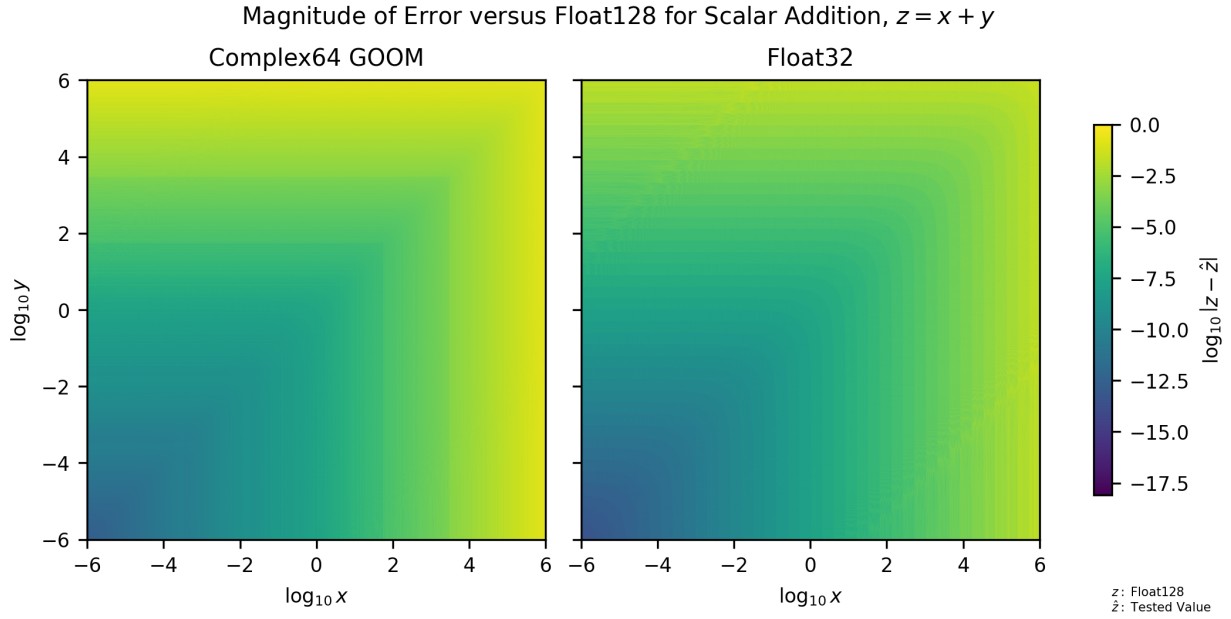

### D.2.7 Magnitude of Errors on Scalar Product/Division

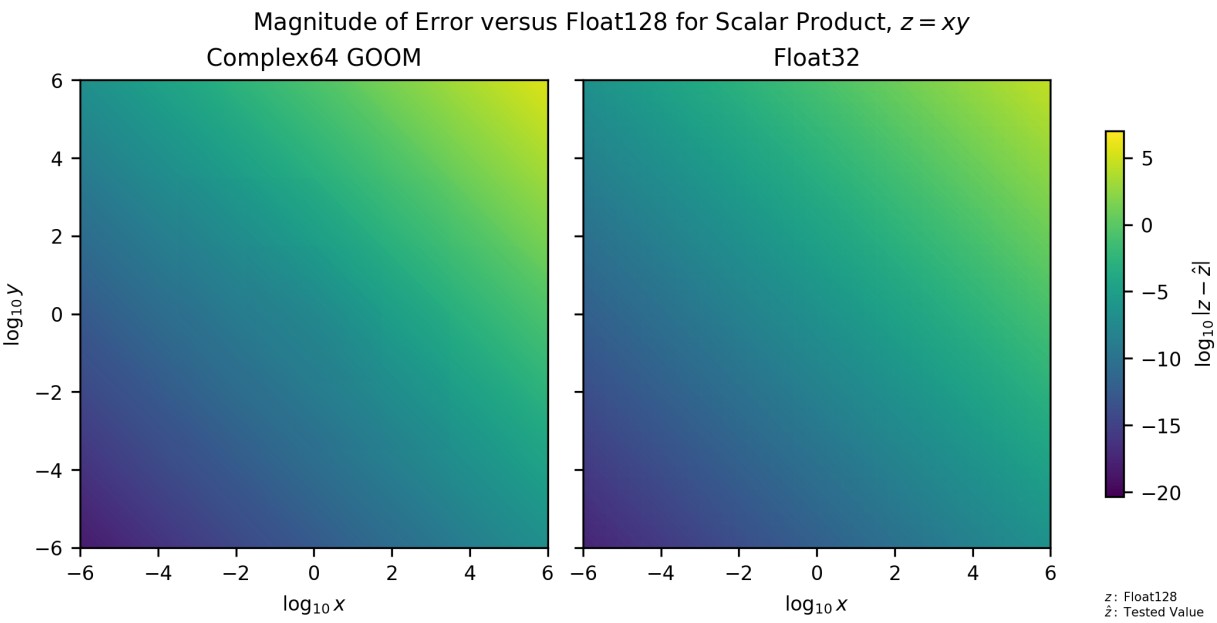

### D.2.8 Normalized Errors on Representative Matrix Product

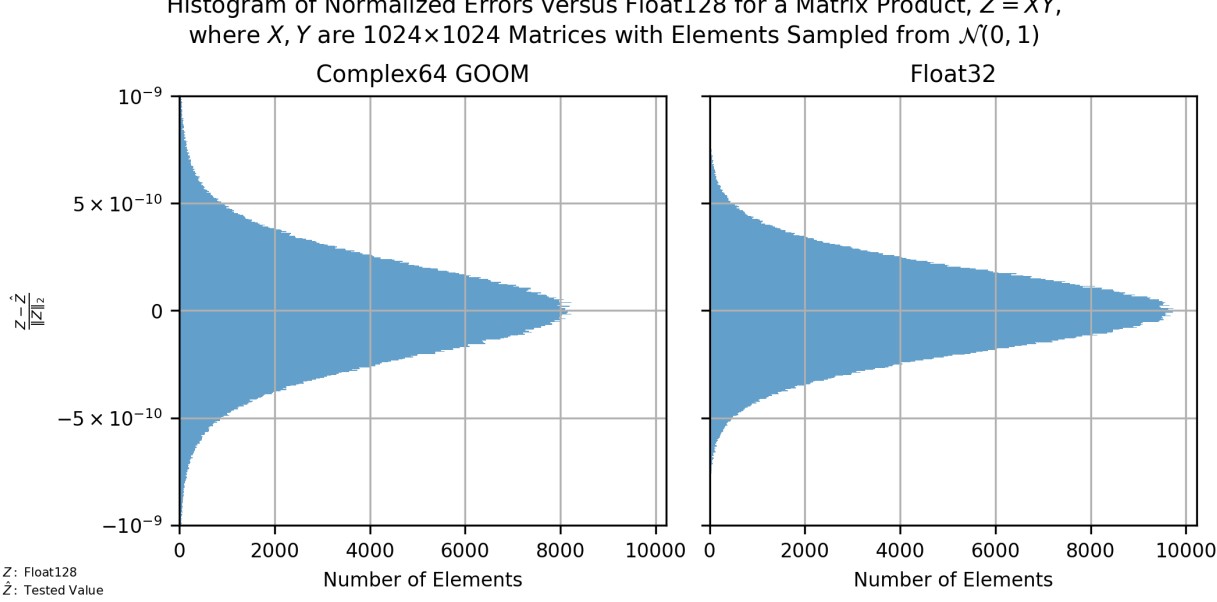

### D.2.9 Execution Times

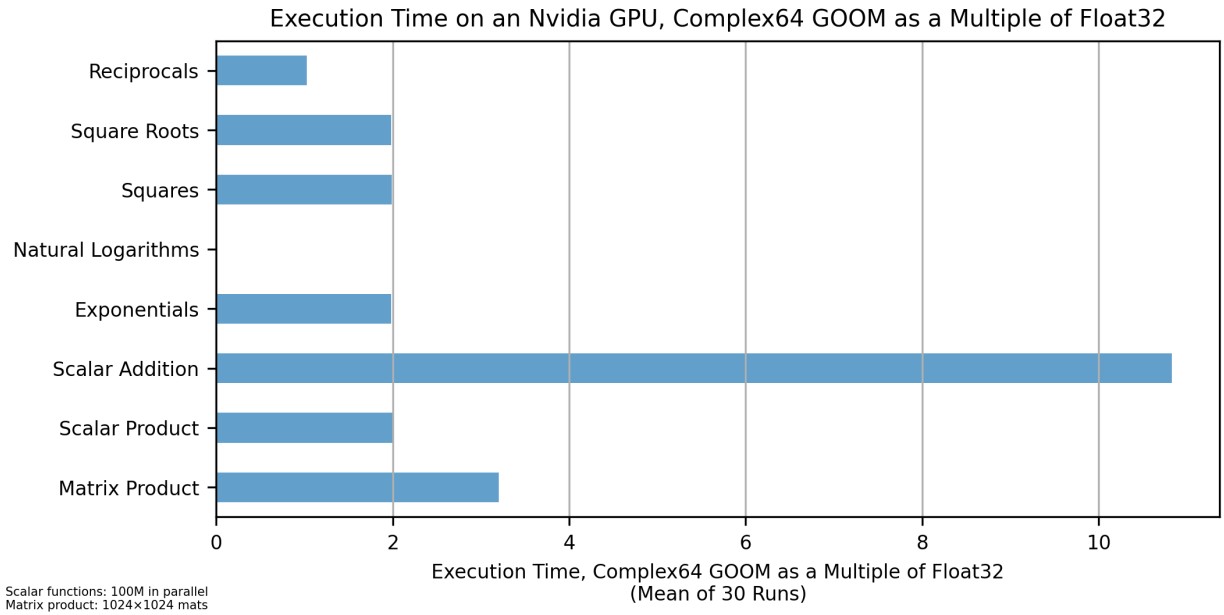

Mean execution times are as reported by `torch.utils.benchmark.Timer`. To the extent possible, we apply transformations in-place, to minimize the impact of new memory allocations on execution time. Complex64 GOOMs are already natural logarithms, so obtaining them incurs no computation.

### D.2.10 Peak Memory Allocated

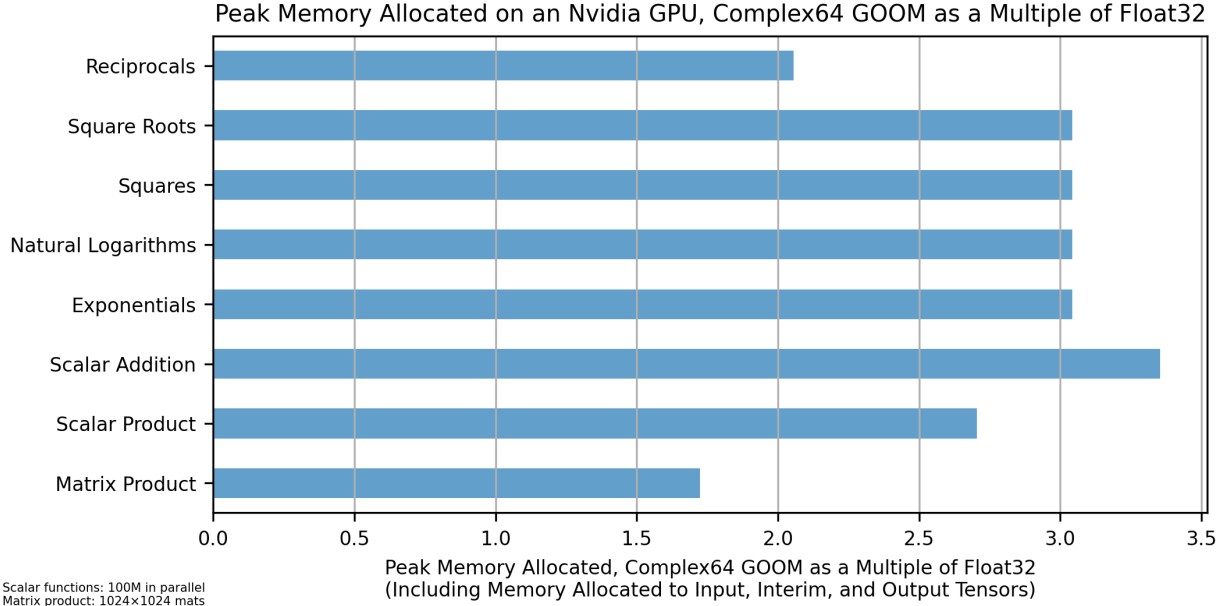

Memory allocation figures are as reported by `torch.cuda.memory.max_memory_allocated`. To the extent possible, we do *not* apply transformations in-place, to allocate memory for input, interim, and output tensors. Memory use by the matrix product over Complex64 GOOMs is for our initial implementation of LMME.

