# OpenReview forum: "Generalized Orders of Magnitude for Scalable, Parallel, High-Dynamic-Range Computation"
_TMLR — Accepted by TMLR_

### Review · Reviewer_djDE · 2025-07-31

**Summary Of Contributions:**

This paper introduces GOOM, an alternative for storing real numbers as complex-valued representations. The real part is used to store the log-magnitude, and the imaginary part is used to store the sign. GOOM extends the dynamic range of traditional floating-point number formats like float32/64. Using GOOM, the authors implement the computation of matrix multiplications, Lyapunov spectra, and neural network training.

**Audience:**

Yes

**Broader Impact Concerns:**

No concerns.

**Claims And Evidence:**

Yes

**Requested Changes:**

* Add further experiments on running time/memory usage/relative error comparisons between GOOM and traditional floating-point numbers.
* Show how GOOM is compared to alternative extended-range formats, such as Posit Arithmetic[2].

[2] Gustafson JL, Yonemoto IT. Beating floating point at its own game: Posit arithmetic. Supercomputing frontiers and innovations. 2017 Apr 25;4(2):71-86.

**Strengths And Weaknesses:**

Strength:
* The proposed method is easy to implement, intuitive, and effective in expanding the dynamic range of traditional floating-point numbers.
* The authors conduct various experiments to test the effectiveness of their method. They also design and implement an efficient parallel algorithm for computing the Lyapunov spectra.

Weakness:
* The idea of using log to handle large number multiplications is not new (e.g., [1]).
* The proposed method is not very efficient, according to my understanding. While it simplifies multiplication, it makes addition much more complicated, considering that taking the exp/log is relatively slow. Additionally, half of the storage (the imaginary part) is used solely to store a binary message (the sign), resulting in a significant waste of memory.
* No running time/memory usage/stability analysis results are reported. It is not clear how the proposed method compares to floating-point numbers in these aspects.

[1] Parhami B. Computing with logarithmic number system arithmetic: Implementation methods and performance benefits. Computers & Electrical Engineering. 2020 Oct 1;87:106800.

---

> ### Author Response · Authors · 2025-08-25
> **Thank you for the careful reading and feedback**
>
> We appreciate the reviewer’s careful reading and constructive requested changes. We have gone through them and made the corresponding alterations to our manuscript, which we believe have improved it.
>
> ---
> **Reviewer Comment:**
> Add further experiments on running time/memory usage/relative error comparisons between GOOM and traditional floating-point numbers.
>
> **Response:**
> We agree with the reviewer that this will strengthen our manuscript significantly. We have added a new 11-page appendix (Appendix D) presenting running time, memory usage, and relative error benchmarks for matrix products, carried out up to the precision limits of Float32 and Float64.
>
> ---
>
> **Reviewer Comment:**
> Show how GOOM is compared to alternative extended-range formats, such as Posit Arithmetic [2].
>
> **Response:**
> We thank the reviewer for raising the comparison to alternative extended-range formats such as Posit arithmetic [2].
>
> In the revised manuscript, we now explicitly note that extended-range formats like Posits can be regarded as special cases of GOOMs. To address the reviewer’s feedback directly, we also discuss Posits in more detail, emphasizing their comparatively limited dynamic range in any practical configuration that can replace conventional floats.
>
> For instance, Table 3 of [2] shows that 64-bit Posits with 4 exponent bits cover magnitudes from roughly:
>
> -  $10^{-299} \approx \exp(-10^{2.8376})$
> - $10^{299} \approx \exp(10^{2.8376})$
>
> By contrast, our 64-bit complex-typed GOOM implementation spans approximately:
>
> - $\exp(-10^{38})\$ to $\exp(10^{38})$
>
> (excluding subnormal floating-point values, which would only further increase the relative advantage of our implementation).
>
> We further clarify that Posits are primarily proposed as replacements for conventional floating-point formats, whereas our implementation of GOOMs is designed to *complement* those formats by extending their representable range in otherwise numerically unstable regimes.
>
> We appreciate the reviewer’s suggestion and have revised the manuscript to highlight this distinction more explicitly.

---

### Review · Reviewer_o1nF · 2025-08-02

**Summary Of Contributions:**

The authors propose a software-level method representing real numbers as complex logarithms (the subset of the complex plane that exponentiates to ℝ), which is inspired from earlier logarithmic number systems. They demonstrate how this allows for numerically stable computation across a larger dynamic range than float32/64, with applications including long matrix product chains such as in Lyapunov spectrum estimation and and training RNNs. The implementation is integrated into PyTorch and exploits GPU parallelism via prefix scans. In addition, the authors provide a novel parallelized algorithm for computing the Lyapunov spectrum of high-dimensional systems.

**Audience:**

Yes

**Broader Impact Concerns:**

/

**Claims And Evidence:**

Yes

**Requested Changes:**

- The authors mention that their approach builds on prior work on logarithmic number systems (LNS), going back to the work by Kingsbury and Rayner (1971). Also the log-sum-exp trick has been standard in this literature. As a non-expert reader, my impression is that the conceptual innovation here to use the multiple of $\pi i$ to represent the sign, however the implications of this change compared to manually representing the sign where not fully obvious to me. Please clarify, and in particular provide a more detailed review of previous and related work on LNS or similar approaches.
- To a reader not familiar with the state of the art in estimating Lyapunov spectra, the introduction and discussion imply that previous approaches where always sequential instead of parallel as per the proposed method. In section 4.2 one reference is given regarding this. Could the authors add a short paragraph giving an overview of previous methods?
- Section 2: What kind of transformations leveraging the structure of the complex plane (e.g., rotations in $\mathbb{C}$) are allowed that would not break exponentiation to $\mathbb{R}$?
- The experiments demonstrate that the computation does not run into over- or underflow, but there is no evidence of the precision of the results. Is it feasible and interesting to add measurements of numerical error for a toy-task example by comparing to analytically computed values? For example, this could be a simple toy task of matrix exponentiation, or comparison of the computed Lyapunov spectrum to an analytically known spectrum.
- The manuscript clearly explains why the logarithmic numbers have a higher dynamic range, and that the representable numbers when using floats as the GOOM exponents are gracefully distributed across the higher range. It is also clear that multiplication will be numerically stable with GOOMs for (very) large numbers, and that the experiments demonstrate the practical success on the example applications (although see the question on precision above). However, it appears that using the system must also come with costs, which are currently not clearly articulated. Firstly, the difficulty of performing addition and in how far it is overcome by the log-sum-exp trick could be explained more clearly, and existing literature studying this trade-off should be mentioned. Secondly, are there operations where the numerical error due to floating point precision will be greater for float+GOOM than for floats themselves? As a non-expert reader I came away from the manuscript with the impression that GOOMs could be a good tool, but that I am not sure where this tool may fail, which is of course detrimental to adoption. Giving clear guidelines or references on such limitations would therefore improve the impact of the paper. I leave the extent to which these questions should be answered by additional analysis or improved discussion to the authors.

**Strengths And Weaknesses:**

## Strengths

- Clear presentation
- Tackles a significant problem of very deep or recurrent architectures trained over long time horizons
- Introduces a parallelized algorithm for computing Lyapunov spectra, based on the author's resetting method
- Implementation in pytorch

## Weaknesses

- Almost no review of previous related literature
- Numerical experiments could be strengthened by demonstrating small numerical error, beyond avoidance of under/overflow
- Clearer delineation of limitations would make the method easier to re-use

Overall, I believe the manuscript makes a significant contribution, is clearly presented, and I did not identify technical issues (though see restriction below). The main weakness that should be addressed is the lack of embedding in the state of previous related literature, since the use of the log domain to avoid float over/underflow is a well-known approach.

Please note that I am not an expert in low-level numerical computing, and have therefore reviewed the manuscript on a less detailed level.

---

> ### Author Response · Authors · 2025-08-25
> **Thank you for the helpful feedback**
>
> We thank the reviewer for their thoughtful and constructive feedback. We have gone through their requested changes, modifying the manuscript accordingly, and we believe this has improved the quality of our manuscript significantly. We address each of these points in turn below.
>
> ---
>
> **Reviewer Comment:**  The authors mention that their approach builds on prior work on logarithmic number systems (LNS)....
>
> **Response:**
> We thank the reviewer for making this suggestion. We have included references to Kingsbury and Rayner (1971) in our manuscript, as well as several other works which combine LNS with deep learning. We have also, at the suggestion of Reviewer djDE, quantitatively compared our implementation of GOOMs to Posits, a related number system. We believe the inclusion of these comparisons has strengthened and contextualized our results in the literature.
>
> ---
>
>
> **Reviewer Comment:**  To a reader not familiar with the state of the art in estimating Lyapunov spectra...
>
> **Response:**
> We thank the reviewer for pointing out this oversight. In response, we have added a short paragraph (the opening paragraph of Section 4.2) that outlines the standard method for computing the largest Lyapunov exponent and clarifies why it cannot be parallelized using an associative scan. A similar reasoning applies to the algorithm for computing the full Lyapunov spectrum.
>
> ---
>
>
> **Reviewer Comment:**  Section 2: What kind of transformations leveraging the structure of the complex plane (e.g., rotations in ℂ) are allowed that would not break exponentiation to **ℝ**?
>
> **Response:**
> We thank the reviewer for raising this subtle question. In general, any transformation that first operates over $\mathbb{C}$ and then maps from $\mathbb{C} \to \mathbb{C}'$ is admissible. As a non-trivial example, one may consider a deep learning model that processes data in $\mathbb{C}$ but includes a final layer mapping from $\mathbb{C} \to \mathbb{C}'$, thereby allowing the data to be scaled and projected into $\mathbb{R}$.
>
> We have added this example to the manuscript, emphasizing that such models can be implemented and trained straightforwardly within our framework. This is possible because we have ensured that backpropagation works seamlessly over $\mathbb{C}$, over $\mathbb{C}'$, and across mappings between $\mathbb{C}' \leftrightarrow \mathbb{R}$, by extending PyTorch’s infrastructure for complex data types. We thank the reviewer for prompting us to clarify and illustrate this capability more explicitly.
>
>
> ---
>
> **Reviewer Comment:**  The experiments demonstrate that the computation does not run into over- or underflow....
>
> **Response:**
> We thank the reviewer for raising this important point.
>
> Direct comparisons against analytic values are not feasible for two of our three representative experiments (chains of matrix products that compound magnitudes toward infinity, and deep learning with parallelized non-diagonal SSMs), since these tasks encounter catastrophic numerical errors under all floating-point formats supported by Nvidia GPUs.
>
> For the remaining experiment—parallel estimation of Lyapunov exponents—we cannot compare directly to analytic values (as these are generally impossible to compute). However, we can benchmark against sequential estimates obtained by standard methods. We have performed this comparison and found the parallel estimates to be similarly accurate. This information was unintentionally omitted from the first draft and is now included in Section 4.2.
>
> The camera-ready version will also provide source code to replicate all experiments, including parallel exponent estimation for all chaotic systems in the dysts dataset (currently 120+ systems). We have also added a new 11-page appendix (Appendix D) reporting running time, memory usage, and relative error of our method versus Float32 and Float64 on matrix products, up to the respective precision limits of those formats.
>
> ---
>
> **Reviewer Comment:**  The manuscript clearly explains why the logarithmic numbers have a higher dynamic range...
>
> **Response:**
> We thank the reviewer for giving us the opportunity to clarify these points.
>
> We can articulate the costs and limitations of our specific implementation, though not for all possible implementations of GOOMs. When compared to Float32 and Float64 (the two floating-point formats supported by Nvidia GPUs) over the relatively small subset of magnitudes they can represent, our implementation of GOOM has longer running times, consumes more memory, and may become less precise as magnitudes grow.
>
> However, where these floating-point formats fail due to overflow or underflow, our implementation of GOOMs behaves fine. In response to the reviewer’s feedback, we have added a new 11-page appendix that quantifies these differences, as described above.

---

### Review · Reviewer_Tz4F · 2025-08-20

**Summary Of Contributions:**

The paper presents Generalized Orders of Magnitude (GOOMs), a complex-logarithmic representation of real numbers designed to enable stable computation across dynamic ranges beyond floating-point limits. The contributions are: (1) formalization of GOOMs and demonstration that floating-point numbers are a special case; (2) implementation of GOOMs in PyTorch with GPU/autograd support; (3) introduction of a selective-resetting method for parallel prefix scans; and (4) demonstration of applications in long chains of matrix products, Lyapunov exponent estimation, and recurrent neural networks.

**Audience:**

Yes

**Broader Impact Concerns:**

This work is primarily technical in nature, focusing on numerical representations and stability in high-dynamic-range computation. As such, it does not raise immediate ethical concerns typical of machine learning applications (e.g., bias, fairness, misuse of generative models).

The main broader-impact considerations relate to:

1. Computational Efficiency and Resource Use: If GOOMs incur significant overhead compared to floating-point arithmetic, widespread adoption could lead to higher energy consumption and reduced efficiency in large-scale ML training. The paper may need to briefly discuss these trade-offs.

2. Accessibility and Usability: Introducing a new numerical representation could affect software and hardware ecosystems. While this work remains at the software-level, readers would benefit from a reflection on whether adoption poses barriers for practitioners unfamiliar with complex arithmetic.

Overall, there are no major ethical risks, but a short broader-impact statement acknowledging the potential implications for compute efficiency, sustainability, and adoption costs would make the submission more complete.

**Claims And Evidence:**

No

**Requested Changes:**

1. Clarify Relation to Prior Work: The paper could more clearly articulate how GOOMs differ from and extend traditional logarithmic number systems (LNS) and log-domain arithmetic methods. At present, the novelty may appear incremental without explicit comparisons and positioning.

2. Expand Empirical Validation: Current experiments are largely synthetic or small-scale. It would be important to include evaluations on real-world machine learning or scientific computing tasks, and to compare against standard stabilization techniques (e.g., gradient clipping, normalization methods in RNNs, extended precision or log-space arithmetic).

3. Performance Benchmarks: Provide more detailed benchmarks quantifying the computational and memory overheads of GOOMs relative to conventional floating-point methods. For example: FLOPs, runtime, GPU utilization, and scalability beyond the toy settings presented.

4. Improve Exposition and Accessibility: Sections on the definition of GOOMs and the selective-resetting method could benefit from clearer intuition, figures, or worked examples earlier in the main text. The appendix is helpful, but the core paper would be more readable if some of that material were integrated.

5. Applications Beyond Demonstrations: If possible, demonstrate GOOMs on a larger-scale deep learning task (e.g., a benchmark language model or transformer setting), or a scientific computing workflow, to highlight practical impact and not just conceptual novelty.

**Strengths And Weaknesses:**

Strengths:

1. Novel idea: Encoding real numbers as complex logarithms seems to be impactful.

2. Breadth of applications: Demonstrated in numerical linear algebra, dynamical systems, and deep learning.

3. Practical implementation: GOOMs are implemented in PyTorch with GPU support and autograd compatibility.

4. Parallelization aspect: Combining GOOMs with prefix scans and selective resetting is technically interesting.

Weaknesses:

1. Positioning vs. prior work: The distinction from logarithmic number systems (LNS) and prior log-domain methods is not sufficiently clear.

2. Empirical validation: Experiments are mostly small-scale or synthetic; comparisons against established stabilization techniques (e.g., normalization, gradient clipping, high-precision arithmetic) are missing.

3. Performance: The compromise implementation of log-matrix-multiplication-exp (LMME) incurs overhead (~2× slower) without thorough benchmarking.

4. Clarity: Parts of the exposition (e.g., selective resetting, definition of GOOMs) are abstract and mathematically heavy, which may limit accessibility.

---

> ### Author Response · Authors · 2025-08-25
> **Thank you for the thorough review and helpful feedback**
>
> We thank the reviewer for their thorough and thoughtful review of our paper. We have carefully gone through their requested changes and made the corresponding adjustments to the manuscript. We believe that this has improved the quality of the manuscript considerably.
>
> ---
>
> ###
> **Reviewer comment:** The paper could more clearly articulate how GOOMs differ from and extend traditional logarithmic number systems (LNS) and log-domain arithmetic methods. At present, the novelty may appear incremental without explicit comparisons and positioning.
>
> **Response:**
> We thank the reviewer for the opportunity to better situate our work within the broader literature. All three reviewers raised this concern, and we agree that our original manuscript lacked sufficient context. In response, we have added references to Kingsbury and Rayner (1971), as well as to several more recent works that combine LNS with deep learning. Additionally, following Reviewer djDE’s suggestion, we have quantitatively compared our implementation of GOOMs with Posits, a related number system. We believe these additions have substantially strengthened our manuscript and better contextualized our results within the existing body of work.
>
> ---
>
> **Reviewer comment:** Current experiments are largely synthetic or small-scale. It would be important to include evaluations on real-world machine learning or scientific computing tasks, and to compare against standard stabilization techniques (e.g., gradient clipping, normalization methods in RNNs, extended precision or log-space arithmetic).
>
> **Response:**
> We thank the reviewer for raising this point. In response, we trained our RNNs within the GOOM framework on a natural language generation task using *The Pile* benchmark dataset (Gao et al., 2020). Specifically, we used a 124M-parameter RNN with a 50,257-token vocabulary and 24 layers, trained on up to 10B tokens, with a standard sequence length of 1024 tokens. The training dynamics are shown in Figure 4. While comparable results can likely be obtained with standard normalization techniques, we emphasize that our approach relies on non-diagonal recurrences computed in parallel via a prefix scan, without any form of stabilization (i.e., no gradient clipping).
>
> ---
>
>
> **Reviewer comment:** Provide more detailed benchmarks quantifying the computational and memory overheads of GOOMs relative to conventional floating-point methods. For example: FLOPs, runtime, GPU utilization, and scalability beyond the toy settings presented.
>
> **Response:**
> We thank the reviewer for this suggestion, which was also raised by all three reviewers. In response, we have added extensive comparisons in Appendix D. We note, however, that GOOMs enable computations (e.g., compound matrix products) far beyond the limits of conventional floating-point arithmetic, which makes direct comparisons challenging. Nevertheless, we provide comparisons with floating-point methods within the shared range of precision wherever possible.
>
> ---
>
> **Reviewer comment:** Sections on the definition of GOOMs and the selective-resetting method could benefit from clearer intuition, figures, or worked examples earlier in the main text. The appendix is helpful, but the core paper would be more readable if some of that material were integrated.
>
> **Response:**
> We thank the reviewer for raising this concern. In response, we have revised the manuscript to improve clarity and overall exposition. Specifically, we now provide clearer explanations early on regarding the distinction between GOOMs and classical floating-point representations, with particular emphasis on the difference between the mathematical formulation of GOOMs and our specific numerical implementation. In addition, we have expanded the discussion of sequential algorithms for computing Lyapunov exponents, highlighting how GOOMs enable orders-of-magnitude speedups over what was previously thought possible. Finally, we have modified our explanation of selective resetting to show explicitly how the standard binary associative transformation (for non-diagonal linear recurrences with biases) compares to the modified transformation, for readability.
>
> ---
>
> **Reviewer comment:** If possible, demonstrate GOOMs on a larger-scale deep learning task (e.g., a benchmark language model or transformer setting), or a scientific computing workflow, to highlight practical impact and not just conceptual novelty.
>
> **Response:**
> We agree with the reviewer. As noted above, we have added a large-scale training run on a natural language generation task (*The Pile*), as well as extensive comparisons between GOOM and floating-point operations in Appendix D, which underlie the majority of scientific computing workflows. We hope that this highlights the practical relevance of our contribution.

---

### Decision · Action_Editor_RjnW · 2025-09-17

**Recommendation:** Accept as is

**Additional Comments:**

This paper is of broad interest and demonstrates good results. It may be in the authors interest to make sure to further communicate that 2/3 examples are only possible with their approach (one reviewer remained a little confused by this even after the rebuttal).

**Audience:**

Yes

**Audience Explanation:**

All three reviewers agree that this paper is of interest to the TMLR audience. In particular, it spans both fundamental computing and machine learning.

**Claims And Evidence:**

Yes

**Claims Explanation:**

Two of the three reviewers said that the claims were supported. The other reviewer focused on the fact that the proposed method does not achieve better results than standard floating-point methods (Appendix D). I went through this point and found that the authors clearly state in their responses to the other two reviewers that for 2/3 of their experiments, floating-point methods don't work. Therefore, while for one of the examples the performance may be similar, the other two examples are only really possible with the GOOM framework. This to is clear evidence to the claim that the GOOM framework is powerful.

---

> ### Author Response · Authors · 2025-09-23
> **Thank you to the reviewers and the AE**
>
> We sincerely thank the reviewers for their careful reading of our paper and for the many constructive comments and suggestions. We believe the manuscript has been significantly improved by addressing these points. We have uploaded the camera-ready version, which now includes a link to a GitHub repository with detailed instructions for reproducing all the experiments in our work. We also extend our gratitude to the Action Editor for their thoughtful evaluation and for the positive final decision.